# Antiproliferative activity of *Grewia villosa* ethyl acetate extract on cervical cancer HeLa cell line: Mechanistic insights through network pharmacology and functional assays approach

**Sally Wambui Kamau[1,2]\***, **Mercy Jepkorir[2]**, **Gilbert Kipkoech[1,2]**, **Inyani John Lino Lagu[3]**, **Wesley Kanda[2]**, **Susan Kibunja[1,2]**, **Rakita Letoluo[2]**, **Shadrack Barmasai[4]**, **Alice Wanyoko[2]**, **Vincent Ruttoh[4]**, **James Kuria[2]**, **Peter Githaiga Mwitari[2]**, **Mathew Piero Ngugi[1]**, **Sospeter Ngoci Njeru** [2,5]\*

**1** Department of Biochemistry, Microbiology and Biotechnology, Kenyatta University, Nairobi, Kenya, **2** Centre for Traditional Medicine and Drug Research (CTMDR), Kenya Medical Research Institute (KEMRI), Nairobi, Kenya, **3** Pan African Union Institute for Basic Science and Technology (PAUSTI), Nairobi, Kenya, **4** Centre for Virus Research (CVR), Kenya Medical Research Institute (KEMRI), Nairobi, Kenya, **5** Centre for Community Driven Research (CCDR), Kenya Medical Research Institute (KEMRI), Kirinyaga, Kenya

\* snjeru@kemri.go.ke, salliekamau@gmail.com

## Abstract

*Grewia villosa* is a plant native to Kenya, with a traditional history among Ambeere people for treating and managing prostate and breast cancers. Previous scientific studies have demonstrated its anti-inflammatory and antioxidant properties. However, a scientific gap exists on the bioactivity of *G. villosa* against cervical cancer, particularly on *in vitro* HeLa cell line model. Additionally, the specific molecular targets and mode of antiproliferative action have not been well elucidated. Therefore, this study sought to investigate the antiproliferative activity, putative targets and mode of action of *G. villosa* using *in vitro* cell culture, molecular biology and *in silico*-based approaches. Antiproliferative analyses were evaluated through MTT assay, cell migration inhibition through *in vitro* scratch assay, and phytochemical profiling through Gas chromatography-mass spectrometry (GC-MS) analysis. Further, putative targets were identified through network pharmacology approach, computationally validated by molecular docking, and functionally through the real-time quantitative polymerase chain reaction (RT-qPCR) method. The *G. villosa* ethyl acetate (GVEA) extract fraction was the most active extract fraction, with IC$_{50}$ of 100.7 µg/mL and a selectivity index of 2.38. Dodecan-2-ylbenzene and 2,6,10-trimethyltetradecane compounds were some notable compounds that can partly be associated with reported antiproliferative activity as they demonstrated strong binding affinity to identified putative targets, including EGFR and AKT1. RT-qPCR analysis functionally confirmed the downregulation of EGFR and AKT1, and the upregulation of tumor protein 53 and Caspase 3 molecular targets, suggesting that GVEA extract indeed perturbs the

**Data availability statement:** All the data and information underlying results presented in this study are available and incorporated within the manuscript and the accompanying supplementary materials provided.

**Funding:** This research partly received support from the KEMRI Internal Research Grant (IRG) funding (KEMRI/IRG/EC0017) to SNN. The funders had no role in study design, data collection and analysis, decision to publish, or preparation of the manuscript.

**Competing interests:** The authors have declared that no competing interests exist.

**Abbreviations:** ADMET, Absorption, Distribution, Metabolism, Excretion and Toxicity; AKT-1, AKT serine/threonine kinase 1; ANOVA, Analysis of Variance; CTMDR, Centre for Traditional Medicine and Drug Research; DMSO, Dimethylsulfoxide; EGFR, Epidermal Growth Factor Receptor; GC-MS, Gas Chromatography-Mass Spectrometry; GLOBOCAN, Global Observatory for Cancer Statistics; GO, Gene Ontology; GVEA, Grewia villosa ethyl acetate extract; HPV, Human papillomavirus; KEGG, Kyoto Encyclopedia of Genes and Genomes Enrichment; LMICs, Low- and middle-income countries; MTT, 3-(4,5-Dimethylthiazol-2-yl)-2,5-Diphenyltetrazolium Bromide; NCBI, National Center for Biotechnology Information; PPI, Protein-protein interaction; RT-qPCR, Real-Time quantitative Polymerase Chain Reaction; SERU, Scientific Ethics and Review Unit; SMILES, Simplified Molecular Input Line Entry Systems; UK, United Kingdom

predicted molecular targets. This study therefore reports the selective antiproliferative properties of the *G. villosa* ethyl acetate extract fraction in a cervical cancer model (HeLa) cell line while at the same time providing putative targets, which is important in shedding light on potential mechanistic basis of its demonstrated antiproliferative activity. This highlights the plant's potential in discovering products and compounds for further investigation on possible application in cervical cancer management and/or treatment.

## 1. Introduction

Cancer is a disease defined by the unregulated proliferation of cells exhibiting immortality. Projections by WHO and American Cancer Society point to an estimated 12 million cancer-related deaths by 2030 and 17.5 million cancer deaths by 2050, respectively [1,2]. Cervical cancer is the fourth most common cancer in women. According to Global Cancer Observatory (GLOBOCAN) estimates, there were an estimated 662,301 new cases of cervical cancer globally, accounting for 3.3% of all new cancer cases and 348,874 deaths [3]. Cervical cancer is primarily caused by persistent infection with high-risk strains of the human papillomavirus (HPV) [4], which often remains asymptomatic. Most infections clear naturally; however, persistent infection with certain HPV strains can lead to precancerous changes in the cervix that eventually progress to invasive cancer [5]. Cervical cancer disproportionately affects low-and middle-income countries (LMICs) because of inadequate healthcare access, few screening programs, limited vaccination drives and treatment options [6,7]. The treatment of cervical cancer depends on the cancer stage and the extent of disease progression. The available treatment strategies may include one or a combination of surgery, radiation, immunotherapy, and chemotherapy [8,9]. Chemotherapy is usually used as an integral part of the standard cervical cancer treatment regimen, either as an adjuvant therapy following surgery to forestall recurrence, in combination with radiotherapy, or as a standalone therapy for locally advanced cervical cancer without PD-1 mutation [10]. Nevertheless, chemotherapy poses significant challenges, including the evolution of chemotherapy resistance, severe adverse reactions, and high treatment costs that negatively impact patient quality of life and overall financial stability. Therefore, research into alternative treatment options is necessary, with natural products and their derivatives, especially those of plant origin, emerging as the "lowest-hanging fruit" with the greatest promise for anticancer drug discovery. Importantly, almost half of all anticancer drugs in clinical trials and close to 60% of currently used drugs have natural product origins [11–15].

Traditional medicinal herbs and plants have long been recognized as valuable sources of therapeutic compounds, offering a rich reservoir of potential anticancer agents [16,17]. Among these plants is *G. villosa* Wild, a member of the *Tiliaceae* family, known locally as "Mubuu" among the Ambeere community of Embu County, Kenya. This plant has been traditionally used in this community for the treatment and management of prostate and breast cancer [18,19]. Some members of the *Grewia*

genus have been shown to have anticancer activity [20]; for example, the methanolic extract of *Grewia asiatica* was shown to have antiproliferative activity against breast cancer MCF-7 cells (IC$_{50}$ 199.5 µg/mL), cervical cancer HeLa cells (177.8 µg/mL), and leukemia HL-60 cells (IC$_{50}$ 53.70 µg/mL) and K-562 cells (IC$_{50}$ 54.90 µg/mL) [21], while the methanolic extract of *Grewia tiliaefolia* inhibited the proliferation of hepatic Hep-2 cells with an IC$_{50}$ of 345 µg/mL [22]. Given the lack of scientific data on the antiproliferative activity of *G. villosa,* we sought to fill this gap. Interestingly, we have demonstrated that *G. villosa* ethyl acetate extract fraction has good antiproliferation activity against the cervical cancer HeLa cell line model (IC$_{50}$ 100.70 µg/mL). Further, the extract inhibited cell migration in a scratch assay. Additionally, we profiled the phytochemicals that can partly or wholly be associated with the reported bioactivity. Moreover, by utilizing these phyto-chemicals, we identified potential targets and modes of action, which we subsequently validated both putatively and func-tionally through molecular docking, and gene expression approaches. Notably, the validated target genes (EGFR, TP53, CASP3 and AKT1) have conserved reported roles in cancer development, progression, and metastasis. Taken together, these exploratory and hypothesis-generating results provide a strong foundation for further studies into the isolation of the compounds for further drug discovery studies against cervical cancer, using multiple cell lines and *in vivo* animal model studies.

## 2. Methods and materials

### 2.1 Collection of plant material

*G. villosa* root barks were collected in Embu County in Central Kenya (0°41'45.9"S, 37°41'10.0"E). The *G. villosa* plant was identified, samples collected, and a voucher specimen (NSN20) was archived at the East African Herbarium. The root bark of *G. villosa* was carefully packaged and taken to the Center for Traditional Medicine and Drug Research (CTMDR) - Kenya Medical Research Institute (KEMRI) for further processing and analysis.

### 2.2 Preparation of the plant extracts

The root barks were thoroughly washed to eliminate any residual soil particles and extraneous materials. Subsequently, the root bark underwent air drying and was ground (Gibbons electric mill – WoodRolfe Road Tollesbury, Essex, UK), followed by storage in an airtight container. One kilogram of the ground root bark material was immersed in 2 L of a 1:1 solvent mixture of Dichloromethane (DCM) and methanol for 72 hours. The resultant mixture was subsequently filtered through Whatman No. 1 filter paper (Schleicher and Schuell Microscience GmbH, Dassel, Germany), and the filtrate was concentrated using a vacuum rotary evaporator (Buchi, Switzerland) maintained at 55 °C to yield total crude extract. This procedure was repeated on the plant marc at intervals of 72 hours until the resultant extracts exhibited a translucent color, indicating the completion of exhaustive extraction. The accumulated extract was then transferred to a separating funnel and partitioned with 400 mL of hexane. This procedure involved vigorous shaking of the mixture, followed by a resting period to facilitate the separation of the less dense hexane extract, which rose to the uppermost layer of the mix. This process was repeated until the introduced hexane stayed colorless even after vigorous shaking, signifying exhaustive extraction of non-polar constituents from the total crude extracts. The hexane extract was subsequently evaporated using a rotary evaporator at 59 °C and appropriately stored. To the remaining total extract, 400 mL of a 1:1 mixture of water and ethyl acetate was added, vigorously shaken, and allowed to rest overnight to form distinct layers. The denser aqueous layer was subjected to lyophilization with a high vacuum freeze dryer (Modulyo Edwards, Crawley, England, UK), while the less dense ethyl acetate layer was concentrated in a vacuum rotary evaporator at 67 °C. The extracted materials were preserved at −20 °C until required for further use [23,24]. 100 mg of each extract was weighed and dissolved in 1 mL of 100% dimethyl sulfoxide (DMSO; Loba Chemie, Mumbai, India) to form a stock solution, which was stored at −20 °C until use. DMSO served as a negative control across all experimental procedures, and its final concentration remained at 0.4% (v/v) or less.

## 2.3 Cell culture

The human cervical cancer cell line (HeLa) and non-cancerous cells derived from the renal tissues of the African Green monkey, *Cercopithecus aethiops* (designated as Vero-ccl 81), were acquired from the American Type Culture Collection (ATCC, USA). The cells were propagated in Minimum Essential Medium (MEM, Gibco, USA), supplemented with 10% Fetal Bovine Serum (FBS, Sigma Aldrich, USA), 1% of a 1M HEPES buffer solution (Sigma Aldrich, USA), 1% of the antibiotic combination Penicillin/Streptomycin (Gibco, USA), 7.5% Sodium Hydrogen Carbonate (Sigma, USA), 1% L-Glutamine (Sigma, USA), and 1.25 mg/L Amphotericin B (Sigma, USA). Cell culture was undertaken in a humidified incubator (Thermo Fisher Scientific, USA), maintained at 5% $CO_2$ and 37°C.

## 2.4 Cell proliferation assay

Cell proliferation was assessed using the MTT assay. In brief, HeLa and Vero cells were cultivated at logarithmic phase in 96-well plates (Biologix, Beijing, China) at a cellular density of $1.0 \times 10^4$ cells/well and incubated overnight to facilitate attachment. Afterward, fresh medium with treatment was added. The HeLa cells underwent an initial screening at a fixed concentration of 200 μg/mL, with the screening threshold set as the extract that inhibited ≥50% of cell proliferation; and this extract would be prioritized for subsequent experimental analysis. Following this, the cells were exposed to a series of concentrations of the prioritized *G. villosa* ethyl acetate extract fraction (GVEA), which ranged from 1000 μg/mL to 15.6 μg/mL for Vero cells and from 400 μg/mL to 6.25 μg/mL for HeLa cells. The absorbance was quantified by a microplate reader (Thermo Scientific, USA) at 570 nm, with 720 nm designated as the reference wavelength. The cytotoxic effects were quantified as the percentage of viable cells in relation to the negative control cells (cells treated with 0.4% DMSO). A standard curve was generated from GraphPad Prism Software (San Diego, California, USA), and the inhibitory concentrations ($IC_{50}$, $IC_{25}$, and $IC_{12.5}$) were subsequently computed through Fit Spline analysis. The experiments were conducted in three technical replicates and were repeated a minimum of three times [23–25]. The formula used to calculate cell viability is:

$$\% \text{ Cell viability } = \frac{\text{Absorbance of the sample} - \text{Absorbance of blank}}{\text{Absorbance of the negative control} - \text{Absorbance of blank}} \times 100$$

## 2.5 Cell morphology assessment

The impact of GVEA extract on HeLa cells was examined by treating cells with three distinct concentrations ($IC_{50}$, $IC_{25}$, and $IC_{12.5}$) for 48 hours. Doxorubicin hydrochloride was used as a positive control at its $IC_{50}$ concentration, while 0.4% DMSO was used as the negative control. Morphological cell alterations were observed and documented utilizing an EVOS™ XL Core Imaging System (Thermo Fisher Scientific, Waltham, MA, USA).

## 2.6 *In vitro* scratch assay

The effect of GVEA extract on the HeLa cell migratory characteristics was assessed through an *in vitro* scratch assay. HeLa cells were seeded in 24-well plates at a concentration of $1.0 \times 10^5$ cells/mL in MEM media and incubated overnight. Subsequently, cells underwent a wash with PBS, after which a scratch was inflicted using a sterile 200 μL pipette tip. An orthogonal line to the scratch was marked using a ruler and a fine-tip permanent marker to demarcate the region for subsequent image analysis. Fresh media with the extract at concentrations corresponding to $IC_{50}$, $IC_{25}$, and $IC_{12.5}$ were introduced, while concurrently, 0.4% DMSO was used as a negative control. Photographs were taken at 0, 24, and 48 hours using an EVOS™ XL Core Imaging System (Thermo Fisher Scientific, Waltham, MA, USA). Scratch areas were analyzed using the Image J software, and the percentage of wound closure was expressed as the percentage change in the normalized measurement divided by the original open area according to the formula:

$$Wound\ closure\ (\%) = [A\,(0) - A\,(t)/\,A\,(0)] * 100$$

where the A(0) is the area at time zero and A(t) is the area after indicated incubation time (24 and 48 hours) [26,27].

### 2.7 Qualitative phytochemical screening

The qualitative phytochemical screening tests of GVEA extract were conducted, and classes of phytochemicals were identified through characteristic color changes according to standard methods. Particularly, sulfuric acid solution and Mayer's reagent test were used to identify alkaloids, water frothing and olive oil emulsion test were used to identify saponins, alkaline reagent (aqueous ammonia solution) test was used to identify flavonoids, Salkowski test was used to identify terpenoids, Keller-Killiani test was used to identify glycosides, ferric chloride and lead acetate test were used to identify tannins, and sodium hydroxide and ferric chloride tests were used to identify phenolics [28–30].

### 2.8 Gas chromatography-mass spectrometry analysis

The method described by Okpako et al., [31] was used, whereby the compounds in GVEA extract were identified by gas chromatography-mass spectrometry (GC-MS) analysis using a Shimadzu GCMS-QP2010SE system (Shimadzu Corporation, Kyoto, Japan) equipped with a low-polarity BPX5 capillary column (30 m × 0.25 mm × 0.25 μm film thickness). The National Institute of Standards and Technology (NIST) mass spectral database was used for compound identification [31].

### 2.9 Network pharmacology analysis

**2.9.1 Compound screening.** Canonical SMILES were generated utilizing the resources available from PubChem (https://pubchem.ncbi.nlm.nih.gov/). Subsequently, the SMILES representations were evaluated for their ADMET (Absorption, Distribution, Metabolism, Excretion, and Toxicity) characteristics employing the Swiss ADME online database (http://www.swissadme.ch/index.php) in conjunction with the pkCSM tool (http://structure.bioc.cam.ac.uk/pkcsm). The parameters obtained from these analytical platforms were utilized to predict the drug-likeness of the compounds in accordance with Lipinski's rules, while applying the following threshold; there should be no more than 5 H-bond donors, there should be less than 10 H-bond acceptors, the molecular weight should not be greater than 500, the calculated LogP should not be greater than 5, and the number of rotatable bonds in a compound should be less than 10. Additionally, compounds' good absorption properties were predicted through indices, including the topological polar surface area (TPSA) threshold of <140 Å$^2$), water solubility threshold in log mol/L of>-6), human intestinal absorption threshold of >30%, and Caco-2 permeability threshold in log Papp ($10^{-6}$ cm/s) of >0.90). Putative good distribution properties were predicted through features including blood–brain barrier (BBB) penetration threshold of logBB > 0.3, and central nervous system (CNS) penetration threshold of logPS > -2. Total clearance was predicted in log mL/min/kg, and the higher the value, the better, while the acceptable predicted threshold for the maximum recommended tolerated dose in humans (MRTD) is ≤ 0.477 log(mg/kg/day) [32–35].

**2.9.2 Identification of the gene targets.** Gene targets pertinent to cervical cancer were extracted from the Online Mendelian Inheritance in Man (OMIM) database, GeneCards database, National Center for Biotechnology Information (NCBI) Gene database, and DISGeNET database, utilizing 'cervical cancer' as the inquiry term. The resultant data sets were consolidated, and redundant targets were eliminated. Furthermore, targets associated with the compounds identified in GVEA extract were predicted using the Swiss Target Prediction (STP), Similarity Ensemble Approach (SEA), and SuperPRED databases. Gene identifiers were sourced from the Universal Protein Resource (UniProt) database. The findings from all databases in each case were integrated, and duplicates removed. An online Venn diagram generator (https://bioinformatics.psb.ugent.be/webtools/Venn/) was employed to ascertain the common genes between the predicted GVEA extract compounds targets and the established cervical cancer genes targets

**2.9.3 Network pharmacological analysis.** The genes characterized as both potential targets of GVEA extract compounds and cervical cancer were entered into the STRING 12.0 database for *Homo sapiens* (https://string-db.org/) with a minimum threshold established at 0.4 to generate a PPI network. The analysis of network topology, predicated on the degree of centrality, was conducted utilizing Cytoscape software version 3.9.1 (National Institute of General Medical Sciences, Bethesda, MD, USA), and the cytoHubba plugin facilitated the identification of the top 30 hub genes. GO and KEGG analyses were conducted utilizing the ShinyGO online platform (http://bioinformatics.sdstate.edu/go/). The following parameters were employed: species = *Homo sapiens*, false discovery rate (FDR) threshold = 0.05, and the number of pathways presented = 20. Enrichment analysis was executed for the molecular function (MF), cellular component (CC), and biological process (BP) ontologies [23].

## 2.10 Molecular docking

The Structured Data File (SDF) format corresponding to the compounds selected after evaluating ADME parameters was obtained from the PubChem database. The molecular structures of the thirty most significant hub genes identified during the network analysis were sourced from the RCSB Protein Data Bank. The preparation of the protein structures was conducted utilizing Discovery Studio 2021, which involved the incorporation of Gasteiger charges, the elimination of water molecules and co-crystallized ligands, as well as the addition of polar hydrogens. The docking of the ligands to the prepared protein structures was performed using the PyRx software version (https://sourceforge.net/projects/pyrx/). A three-dimensional grid box delineating the active site of each protein was established. The complexes exhibiting the lowest binding energies were subsequently visualized using Discovery Studio 2021, generating two-dimensional interaction diagrams [36].

## 2.11 Gene expression analysis

HeLa cells were cultured in T-75 culture flasks. The culture medium was substituted with fresh media containing GVEA extract at $IC_{50}$ concentration for 48 hours. Total RNA was extracted utilizing the Pure Link RNA Kit (Thermo Fisher Scientific, Waltham, MA, USA), Complementary DNA (cDNA) was synthesized employing the FireScript RT cDNA Synthesis Kit (Solis BioDyne, Tartu, Estonia) and RT-qPCR was conducted utilizing the Luna Universal qPCR Master Mix (New England Biolabs, Ipswich, MA, USA) as per the instructions provided by the manufacturer. The primers used were purchased from (Macrogen Europe, Seoul, South Korea) and the sequences are as shown in (Table 1) below. The tested genes' mRNA levels were quantified using the $2^{-\Delta\Delta Ct}$ method, and GAPDH and β-actin used as internal reference controls [37,38].

## 2.12 Statistical analysis

GraphPad prism 8.0 (San Diego California, USA) was used for statistical analysis, and comparison between means was analyzed using ANOVA, one-way analysis followed by Tukey and Dunnett's post hoc test. Statistically significant

**Table 1. Forward and reverse primers used for RT-qPCR.**

| GENES | FORWARD PRIMERS | REVERSE PRIMERS |
|---|---|---|
| TP53 | CTTCGAGATGTTCCGAGAGC | GACCATGAAGGCAGGATGAG |
| EGFR | TCTGGAAGTACGCAGACGCC | TGGGAGACTAAAGTCAGACAGTG |
| CASPASE 3 | CAAAGAGGAAGCACCAGAACCC | GGACTTGGGAAGCATAAGCGA |
| AKT1 | CCATCTGTCACCAGGGGCTT | ATAGCCACGTCGCTCATGGT |
| GAPDH | AGACAGCCGCATCTCTTG | TGACTGTGCCGTTGAACTTG |
| β-ACTIN | GCCAACTTGTCCTTACCCAGA | AGGAACAGAGACCTGACCCC |

differences in comparison to the negative control (0.4% DMSO) are denoted by \*$p < 0.05$, \*\*$p < 0.01$, \*\*\*$p < 0.001$, \*\*\*\*$p < 0.0001$. The data is presented as the mean $\pm$ SD.

## 3. Results

### 3.1 Antiproliferative and cytotoxic activity of *G. villosa* extracts

To assess the antiproliferative properties of *G. villosa* extracts, both the crude extract and its fractions were initially subjected to screening against HeLa cells at a fixed concentration of 200 µg/mL. The ethyl acetate extract was subsequently prioritized for further experimentation due to its good activity, having passed our set threshold of inhibiting cell proliferation by ≥ 50% at the set fixed concentration (Fig 1A). The antiproliferative effect of the ethyl acetate extract was further investigated in a concentration dilution assay against HeLa cells (to determine the half maximal inhibitory concentration ($IC_{50}$)), as well as against non-cancerous Vero cells (to evaluate the selectivity/safety and the half maximal cytotoxic concentration ($CC_{50}$)), using the MTT assay (Fig 1B and 1C). Interestingly, *G. villosa* ethyl acetate (GVEA) extract exhibited the highest inhibition of the proliferation of cervical cancer HeLa cells in a dose-dependent manner over 48 hours. Furthermore, the extract exhibited reduced cytotoxicity to non-cancerous Vero cells compared to cervical cancer HeLa cells. This suggests that GVEA extract is selective in its activity. Additionally, we evaluated the cytotoxicity of doxorubicin hydrochloride (the positive control) on Vero cells (S1 Fig in S1 File.) and its antiproliferative activity against HeLa cells (S2 Fig in S1 File.). As expected, the antiproliferative activity with doxorubicin hydrochloride was higher than with GVEA extract.

### 3.2 Determination of the selectivity index (SI) of *G. villosa* ethyl acetate extract

The selectivity indices were calculated using the $CC_{50}$ (cytotoxic concentration that results in the inhibition of 50% of non-cancerous Vero cells) and the $IC_{50}$ (inhibitory concentration that leads to the inhibition of proliferation of 50% of cervical cancer HeLa cells). The determination of the $CC_{50}$ and $IC_{50}$ was performed by using the data from (Fig 1B and 1C). GVEA extract exhibited an $IC_{50}$ value of 100.7 µg/mL and a $CC_{50}$ value of 240.6 µg/mL. The positive control (doxorubicin hydrochloride) $IC_{50}$ of 1.45 µg/mL and a $CC_{50}$ of 4.51 µg/mL. The bioactivities ($IC_{50}$ and $CC_{50}$) of doxorubicin hydrochloride (which is a pure compound) were higher than the bioactivities recorded for GVEA extract (Table 2). When assessing selectivity, positive control showed the highest activity and selectivity for cancer cells, with a SI of 3.12, while GVEA

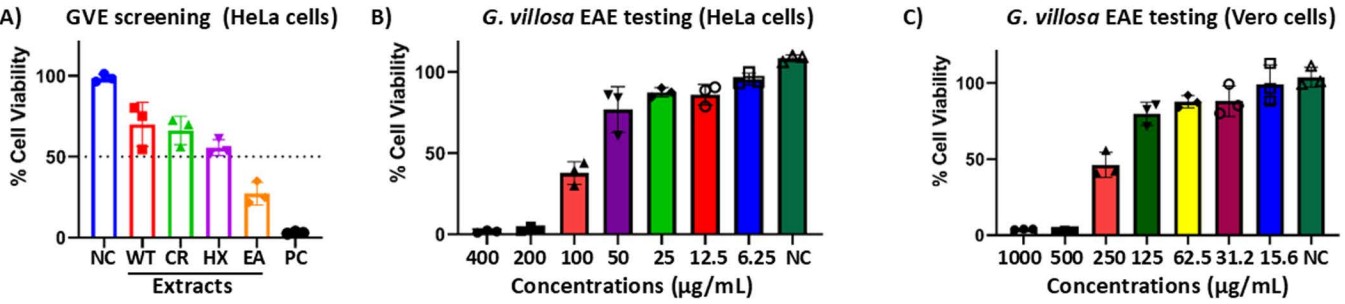

**Fig 1. *G. villosa* ethyl acetate extract selectively inhibits the proliferation of cervical cancer HeLa cells.** (**A**) Initial screening of *G. villosa* extracts. HeLa cells were treated with crude, hexane, ethyl acetate, and water extracts at a fixed concentration of 200 µg/mL. *G. villosa* ethyl acetate extract met our set threshold by inhibiting the proliferation of HeLa cells by ≥ 50% and was, therefore, prioritized for subsequent experimental procedures. (**B – C**) Concentration dilution study of GVEA extract to facilitate calculation of $IC_{50}$ values for HeLa cells (**B**), and $CC_{50}$ values for Vero cells (**C**). The experiments were undertaken in three technical replicates and at least three biological replicates. NC: negative control (0.4% DMSO); WT: water partition extract fraction; CR: crude/total extract; HX: hexane partition extract fraction; EA: ethyl acetate partition extract fraction; PC: positive control (doxorubicin hydrochloride).

extract exhibited a SI of 2.38 (Table 2). It is important to note that an extract or a compound is considered selective if it has a SI of >1 [39].

### 3.3 Effects of GVEA extract on cell morphology

The GVEA extract was used to examine the morphological alterations in cervical cancer HeLa cells after exposure to the extract. Treatment with GVEA extract at $IC_{50}$, $IC_{25}$, $IC_{12.5}$, along with the positive control (2 µg/mL) and negative control (0.4% DMSO), was undertaken for 48 hours and documented by microscopy. The positive control and GVEA extract (at $IC_{50}$) exhibited the greatest alterations on cellular morphology, characterized by cell shrinkage, detachment, and cell death (Fig 2).

### 3.4 Effects of GVEA extract on cell migration

The role of GVEA extract in affecting the HeLa cells' migration patterns was investigated using the *in vitro* scratch assay. The HeLa cells were treated with GVEA extract at $IC_{50}$, $IC_{25}$, and $IC_{12.5}$ as well as with negative control, and microscopically monitored at 0, 24, and 48 hours (Fig 3). Interestingly, GVEA extract inhibited cellular migration in both time and concentration-dependent manner. The $IC_{50}$ concentration had the most remarkable inhibitory effect, compared to the negative control.

### 3.5 Qualitative phytochemical screening of GVEA extract

Qualitative phytochemical screening was undertaken following standard methods, and the results show that tannins and phenols were highly abundant, alkaloids were moderately abundant, and saponins and terpenoids were present. These are groups of phytochemicals that can partly or wholly be associated with the demonstrated bioactivity of GVEA extract. Glycosides and flavonoids were, however, not detected (Table 3), indicating absence or presence below our qualitative detection limits.

**Table 2. Selectivity indices of GVEA extract and doxorubicin hydrochloride.**

| Extract | $IC_{50}$ (µg/mL)±SD | $CC_{50}$ (µg/mL)±SD | Selectivity Index |
|---|---|---|---|
| Ethyl acetate | 100.70±4.99 | 240.67±6.81 | **2.38** |
| Doxorubicin hydrochloride | 1.45±0.24 | 4.52±1.24 | **3.12** |

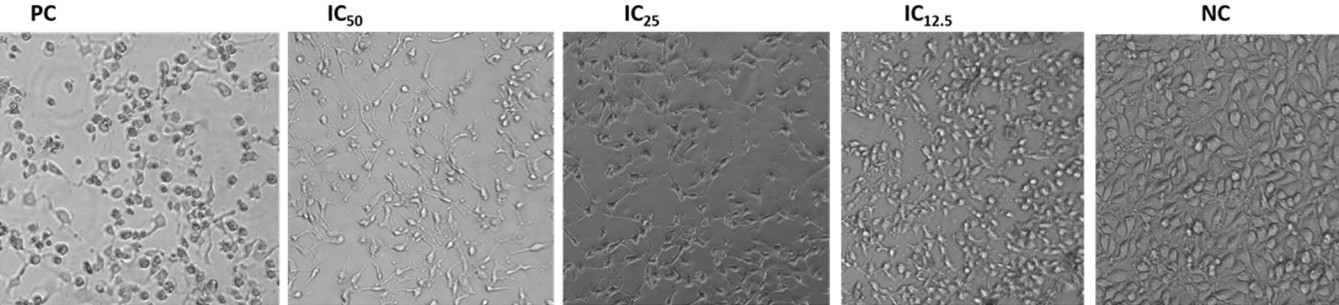

**Fig 2. Morphological changes of cells after exposure to the GVEA extract at different concentrations.** Treatment with positive control (2 µg/mL) and GVEA extract (at $IC_{50}$) exhibited morphological alterations, characterized by cell shrinkage, detachment, and cell death, while the negative control exhibited normal cell growth. Three technical and three biological replicates were used. PC: positive control (doxorubicin hydrochloride); NC: negative control (0.4% DMSO); IC: inhibitory concentration.

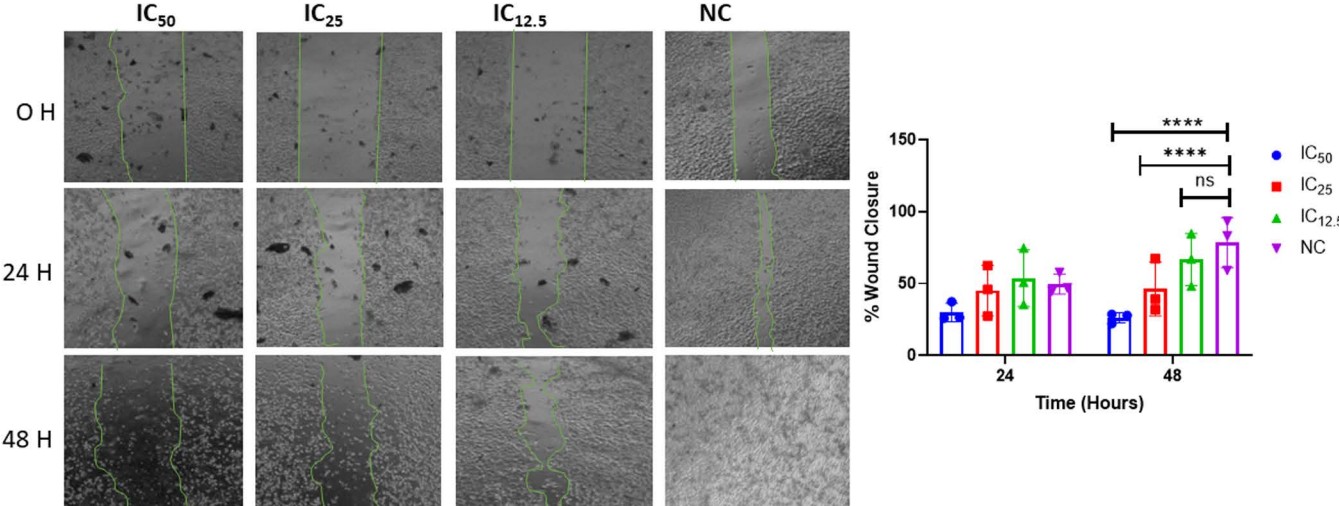

**Fig 3. GVEA extract inhibited cell migration in a concentration and time-dependent manner.** Images on the left show the HeLa cells' wound closure after 0-, 24-, and 48-hour treatment with GVEA extract and the negative control. The graph on the right shows the percentage representation of the wound closure. Three technical and three biological replicates were used. NC: negative control (0.4% DMSO); H: hours; IC: inhibitory concentration; p values, ****: $p < 0.0001$; ns: $p = 0.21$.

## 3.6 Characterization of GVEA extract compounds through GC-MS analysis

To ascertain the distinct chemical constituents present within the active GVEA extract, a GC-MS analysis was undertaken, and 14 peaks were detected (Fig 4). However, the peak for eicosane was repeated four times, and that for tetratetracontane was repeated twice. Therefore, a total of 9 compounds were identified by GC-MS. Eicosane was the most abundant compound with 29.58% total area, followed by Tetratetracontane with 21.95% total area and Octacosane with 11.85% total area (Table 4).

## 3.7 In silico

### 3.7.1 Prediction of physicochemical and pharmacokinetic properties of GVEA extract compounds.
The compounds identified in the GVEA extract were evaluated for their physicochemical and pharmacokinetic properties to help establish drug-likeness using *in silico* ADMET prediction tools. Six out of the nine identified compounds met the

**Table 3. Phytochemical screening results of *G. villosa* ethyl acetate extract.**

|  | Ethyl acetate |
|---|---|
| Alkaloids | *** |
| Saponins | ** |
| Flavonoids | − |
| Terpenoids | − |
| Glycosides | − |
| Tannins | *** |
| Phenols | *** |

**Key:** ***= highly abundant; **= moderately present; - = undetectable quantities

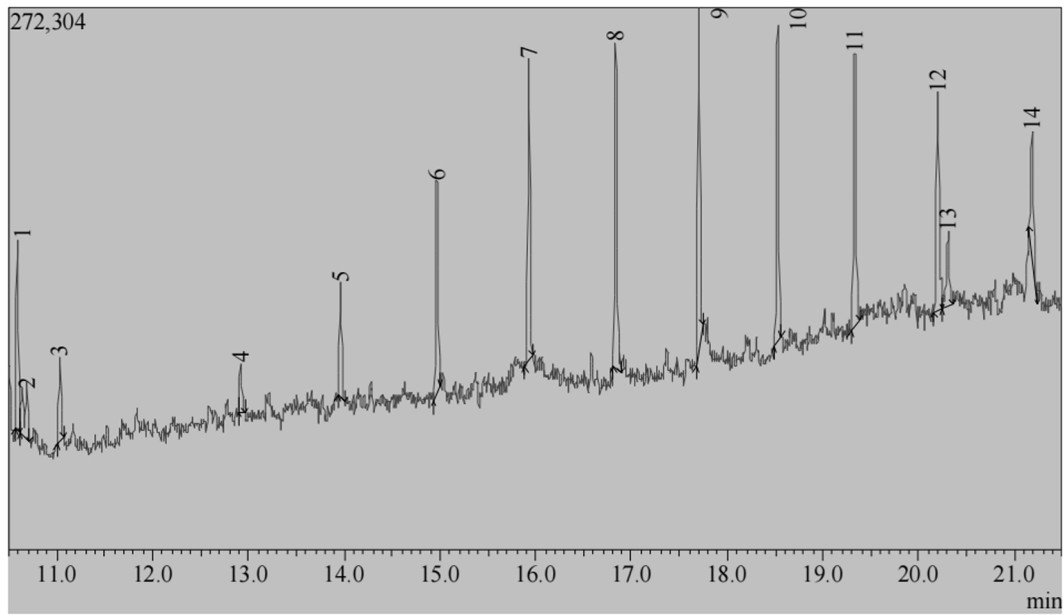

**Fig 4. GC-MS Chromatogram of compounds contained in the G. villosa ethyl acetate extract.** Fourteen compound peaks were identified from G. villosa ethyl acetate extract fraction.

**Table 4. Compounds identified from GVEA extract by GC-MS analysis.**

| Peak | Retention Time | %Area | Compound Identified | IUPAC NAME | Molecular Formula | Molecular Weight (g/mol) | Classification |
|---|---|---|---|---|---|---|---|
| 1 | 10.587 | 6.73 | Glutaric acid, di(1-phenylpropyl) ester | Bis(1-phenylpropyl) pentanedioate | $C_{23}H_{28}O_4$ | 368.5 | Fatty acid ester |
| 2 | 10.693 | 3.57 | Phenol, 2-methyl-4-(1,1,3,3-tetramethylbutyl)- | 2-methyl-4-(2,4,4-trimethylpentan-2-yl)phenol | $C_{15}H_{24}O$ | 220.35l | Phenol |
| 3 | 11.028 | 3.16 | Benzene, (1-methylundecyl)- | Dodecan-2-ylbenzene | $C_{18}H_{30}$ | 246.4 | Aromatic Hydrocarbon |
| 4 | 12.917 | 1.58 | Tetradecane, 2,6,10-trimethyl- | 2,6,10-trimethyltetradecane | $C_{17}H_{36}$ | 240.5 | Hydrocarbon |
| 5 | 13.963 | 3.83 | Eicosane | Icosane | $C_{20}H_{42}$ | 282.5 | Hydrocarbon |
| 6 | 14.966 | 7.92 | Docosane | Docosane | $C_{22}H_{46}$ | 310.6 | Hydrocarbon |
| 7 | 15.926 | 10.23 | Tricosane | Tricosane | $C_{23}H_{48}$ | 324.6 | Hydrocarbon |
| 8 | 16.837 | 11.59 | Eicosane | Icosane | $C_{20}H_{42}$ | 282.5 | Hydrocarbon |
| 9 | 17.703 | 11.85 | Octacosane | Octacosane | $C_{28}H_{58}$ | 394.8 | Hydrocarbon |
| 10 | 18.526 | 11.11 | Tetratetracontane | Tetratetracontane | $C_{44}H_{90}$ | 619.2 | Hydrocarbon |
| 11 | 19.329 | 10.84 | Tetratetracontane | Tetratetracontane | $C_{44}H_{90}$ | 619.2 | Hydrocarbon |
| 12 | 20.197 | 9.42 | Eicosane | icosane | $C_{20}H_{42}$ | 282.5 | Hydrocarbon |
| 13 | 20.305 | 3.43 | Decanedioic acid, bis(2-ethylhexyl) ester | bis(2-ethylhexyl) decanedioate | $C_{26}H_{50}O_4$ | 426.7 | Fatty acid ester |
| 14 | 21.182 | 4.74 | Eicosane | Icosane | $C_{20}H_{42}$ | 282.5 | Hydrocarbon |

desired criteria for drug-likeness, including predicted blood-brain barrier permeability, predicted inhibition of cytochrome enzymes 2D6 and 3A4, compliance with Lipinski's rule of five (no more than two violations), and a topological surface area of less than 140 Å$^2$, and their predicted ADMET properties are shown in (Table 5). Total clearance, a measure of drug elimination from the body, is presented as log (mL/min/kg), and all compounds had predicted values >0.5. AMES toxicity, a test for mutagenicity, was predicted, but none of the compounds were predicted to be mutagenic. The maximum tolerated dose (MTD) in humans was predicted, and for a compound, an MTD ≤ 0.477 Log (mg/kg/day) is considered low, while a value ≥ 0.477 is considered high, and only compounds 1, 2 and 9 had a high predicted MTD (Table 5). The potential for hepatotoxicity (drug-induced liver injury) was predicted, and none of the nine compounds had that likelihood. The inhibition of the hERG1 (encoded by the human ether-a-go-go gene) potassium channel, which is primarily expressed in the heart, is a significant safety concern for drug development, as it can lead to long QT syndrome and potentially fatal cardiac arrhythmias. In this study, none of the compounds in GVEA extract were predicted to inhibit hERG1. However, all compounds except compounds 1, 3, and 9 were predicted to inhibit hERG2.

### 3.7.2 Prediction of compound and cervical cancer genes.

Target genes associated with two compounds from GVEA extract (Dodecan-2-ylbenzene and 2,6,10-trimethyltetradecane) and doxorubicin hydrochloride (as positive control) were predicted from three distinct databases, specifically the SEA database (35), the SuperPRED database (312), and the STP database (286). The identified targets were compiled together, duplicates removed to yield a total of 439 compound target genes. The cervical cancer target genes were predicted from four databases: the OMIM database (120), the NCBI database (1759), the DisGENET database (1817), and the Gene Cards database (6406). Targets from these four databases were aggregated, duplicates removed to yield a total of 6927 cervical cancer-associated gene targets (S1 sheet). Intersection of compound targets and cervical cancer targets by Venn diagrams led to identification of 172 common key genes (Fig 5A–5B).

### 3.7.3 Construction of protein-protein interaction (PPI) network.

A PPI network was generated by STRING database, and it contained 172 nodes and 3179 edges, with an average node degree of 25.7 and an average local clustering coefficient of 0.518, with an anticipated edge count of 1472 with a p-value < 1.0e-16. Subsequently, the network was exported to Cytoscape for the purposes of visualization and analysis, utilizing the Cytohubba plugin to identify the top 30 hub genes based on centralities (betweenness, degree, closeness, maximal clique coefficient, and eccentricity) (Fig 5C).

### 3.7.4 Kyoto encyclopedia of genes and genomes (KEGG) and gene ontology (GO) analysis.

KEGG pathway analysis predicted 183 pathways that exhibited significant enrichment, and the top 20 pathways (-LogP) are shown, whereby, PI$_3$K-AKT-mTOR pathway was of particular interest due to having numerous protein targets identified within the PPI network, which have essential roles in cancer evolution and progression (Fig 5D). The common key genes exhibited substantial enrichment across 1927 Gene Ontology (GO) terms, encompassing 1637 terms related to biological processes, 160 terms associated with molecular functions, and 124 terms about cellular components (p < 0.05). The most prominent GO terms within biological processes included those associated with responses to organic cyclic compounds, responses to lipids, and responses to organonitrogen compounds. In the realm of cellular components, the top GO terms comprised those associated with the membrane raft, membrane microdomain, and receptor complex. The top GO terms related to molecular functions included aminopeptidase activity, nuclear receptor activity, and ligand-activated transcription factor activity (Fig 5E–5G).

## 3.8 Molecular docking analysis

To validate the top 30 hub genes as core targets for the GVEA extract compounds, we computationally evaluated the binding affinity of two GVEA extract compounds (Dodecan-2-ylbenzene and 2,6,10-trimethyltetradecane) as well as doxorubicin hydrochloride (as a positive control). A binding energy value of less than 0 kcal/mol signifies that a ligand and a receptor exhibit spontaneous binding. However, a binding affinity of less than −5.0 kcal/mol denotes a strong

Table 5. In silico prediction for physicochemical and pharmacokinetic properties of GVEA extract compounds.

| Compounds | MW | MOL LOGP | HA | HD | TPSA | LP (rules violated) | ABSORPTION Water solubility | Caco2 permeability | Intestinal absorption (human) | DISTRIBUTION BBB | CNS permeability | METABOLISM CYP1A2 inhibitor | CYP2C19 inhibitor | CYP2C9 inhibitor | CYP2D6 inhibitor | CYP3A4 inhibitor | EXCRETION Total Clearance | Renal OCT2 substrate | TOXICITY AMES toxicity | Max. tolerated dose (human) | hERG I inhibitor | hERG II inhibitor | Hepatotoxicity |
|---|---|---|---|---|---|---|---|---|---|---|---|---|---|---|---|---|---|---|---|---|---|---|---|
| 1 | 220.356 | 4.12 | 1 | 1 | 20.23 | No,0 | -4.898 | 1.649 | 90.819 | Yes 0.46 | -1.59 | Yes | No | No | No | No | 0.779 | No | No | 0.754 | No | No | No |
| 2 | 246.438 | 6.57 | 0 | 0 | 0 | Yes,1 | -7.566 | 1.495 | 92.756 | Yes 0.88 | -1.158 | Yes | Yes | No | No | No | 1.727 | No | No | 0.874 | No | Yes | No |
| 3 | 240.475 | 6.68 | 0 | 0 | 0 | Yes,1 | -8.166 | 1.413 | 92.811 | Yes 0.925 | -1.383 | Yes | No | No | No | No | 1.575 | No | No | 0.238 | No | No | No |
| 4 | 310.61 | 7.82 | 0 | 0 | 0 | Yes,1 | -8.473 | 1.369 | 88.984 | Yes 1.052 | -1.089 | Yes | No | No | No | No | 2.07 | No | No | -0.101 | No | Yes | No |
| 5 | 324.637 | 8.03 | 0 | 0 | 0 | Yes,1 | -8.338 | 1.137 | 88.64 | Yes 1.071 | -1.035 | Yes | No | No | No | No | 2.105 | No | No | -0.144 | No | Yes | No |
| 6 | 394.772 | 9.06 | 0 | 0 | 0 | Yes,1 | -7.085 | 1.114 | 86.922 | Yes 1.166 | -0.762 | Yes | No | No | No | No | 2.119 | No | No | -0.284 | No | Yes | No |
| 7 | 619.204 | 11.92 | 0 | 0 | 0 | Yes,2 | -3.232 | 1.043 | 81.424 | Yes 1.468 | 0.111 | No | No | No | No | No | 2.482 | No | No | 0.32 | No | Yes | No |
| 8 | 282.556 | 7.38 | 0 | 0 | 0 | Yes,1 | -8.59 | 1.371 | 89.671 | Yes 1.014 | -1.199 | Yes | No | No | No | No | 1.998 | No | No | -0.014 | No | Yes | No |
| 9 | 426.682 | 5.39 | 4 | 0 | 52.6 | Yes,1 | -5.453 | 1.352 | 91.099 | No -0.202 | -2.595 | No | No | No | No | No | 1.916 | No | No | 0.628 | No | No | No |

**Key:** MW (Molecular Weight), HA (Hydrogen Acceptor), HD (Hydrogen Donator), MolLogP (Molecular Lipophilicity), BBB (Blood Brain Barrier penetration as predicted through LogBB), CYP2D6 (Cytochrome P450 2D6), CYP3A4 (Cytochrome P450 3A4), TPSA (Topological Surface Area Å²), LR (Lipinski's rule of five violations) and CNS (Central Nervous System permeability as predicted through LogPS).Compounds are: **1**- (2-methyl-4-(2,4,4-trimethylpentan-2-yl)phenol), **2**-(Dodecan-2-ylbenzene), **3**-(2,6,10-trimethyltetradecane), **4**-(Docosane), **5**-(Tricosane), **6**-(Octacosane), **7**-(Tetratetracontane), **8**-(Eicosane) and **9**-(bis(2-ethylhexyl) decanedioate).

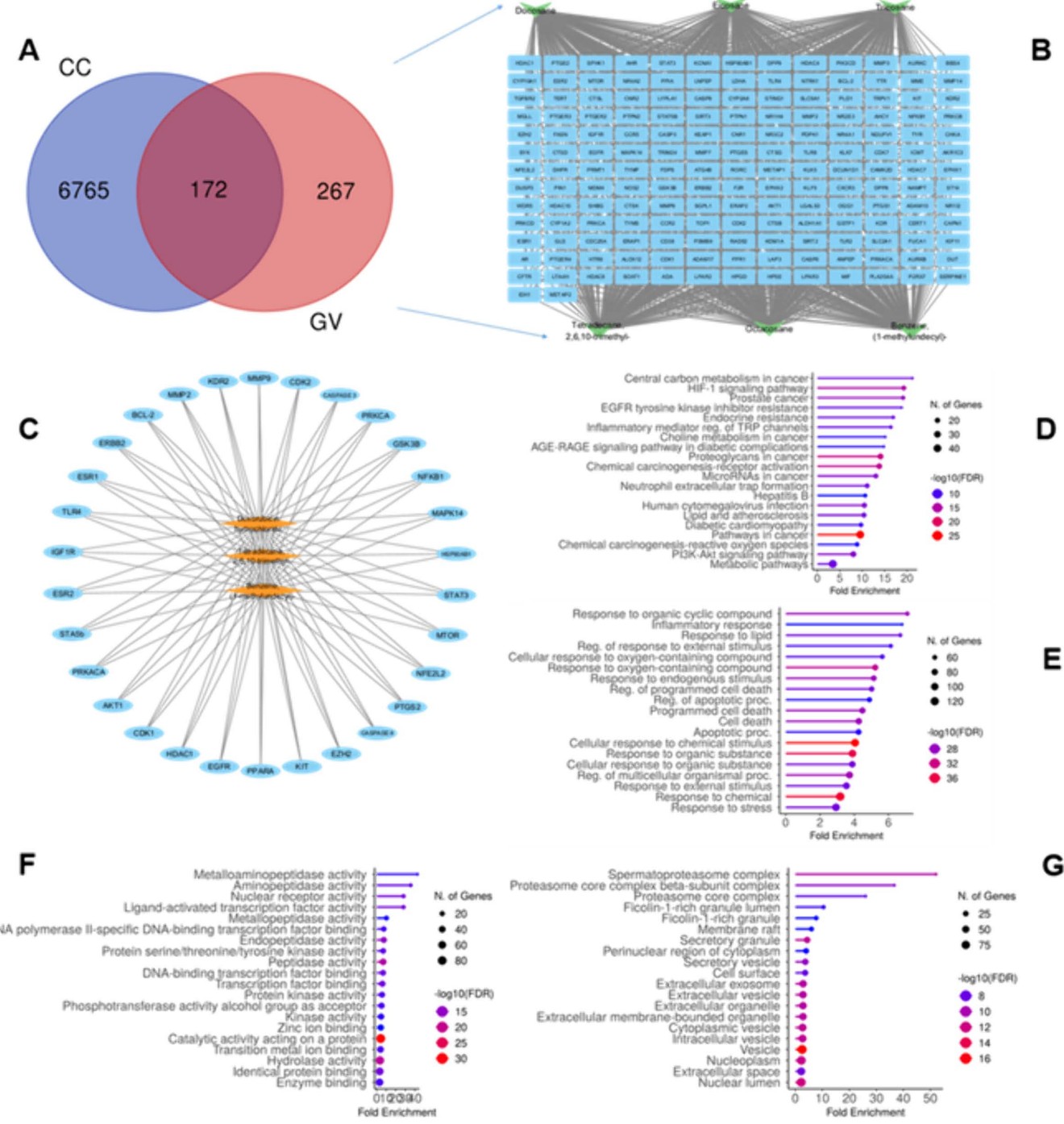

**Fig 5. Prediction of common key target genes, biological functions, and molecular pathways potentially deregulated by GVEA extract compounds. (A)** Venn diagram intersecting GVEA extract compound target gene (GV) and cervical cancer target genes (CC); **(B)** Grid representation of the 172 common key targets genes with the 6 prioritized compounds (refer section 3.7.1); **(C)** Top 30 hub genes with the two GVEA extract compounds and doxorubicin hydrochloride (as a positive control) targeting them; **(D)** KEGG pathway analysis showing the top 20 putative pathways; **E-G)** Gene Ontology analysis showing biological processes **(E)**, molecular functions **(F)** and cellular components **(G)**. The x-axis illustrates the numerical representation of genes associated with each Gene Ontology (GO) term, whereas the y-axis signifies the enriched GO terms. The order of significance is systematically organized from the uppermost to the lowermost section and classified based on –log10 (False Discovery Rate).

binding affinity between the ligand and the receptor, while a binding affinity of less than −7.0 kcal/mol indicates an exceptionally strong binding affinity between the ligand and the receptor [40]. Our 3 tested compounds exhibited a binding affinity of ≤ −4 kcal/mol (Fig 6). For example, the docking configurations of 2,6,10-trimethyltetradecane in association with AKT1 and EGFR are illustrated in (Fig 7). The amino acids implicated in the interaction between EGFR and 2,6,10-trimethyltetradecane encompass Van der Waals forces on GLY 796, ILE 744, THR 854, ASP 855, CYS 775, LEU 858, ARG 176, and GLN 791, as well as Alkyl, Pi-Alkyl, and Pi-Sigma interactions involving LEU 844, MET 793, LEU 792, ALA 743, VAL 726, MET 790, LYS 745, LEU 788, MET 766, and LEU 777. The amino acids participating in the binding of 2,6,10-trimethyltetradecane with AKT1 include Van der Waals forces on VAL 271, THR 211, ASP 292, TYR 263, and SER 205, along with Alkyl, Pi-Alkyl, and Pi-Sigma interactions involving TYR 272, LEU 210, LEU 264, VAL 270, TRP 80, and LYS 268.

### 3.9 Gene expression analysis of GVEA extract-treated HeLa cells

To empirically and functionally validate our network pharmacology and molecular docking predictions, we used RT-qPCR analysis on four target genes; 2 genes (AKT1 and EGFR) identified as top hub genes and tested by molecular docking (binding affinity ≤ −6 kcal/mol) and 2 genes (TP53 and CASP3) selected for their canonical role in cancer development and progression. Importantly, CASP3 is a member of the predicted top 30 hub genes, and all four are involved in the complex interplay of cancer development and progression. The threshold cycle ($C_t$) values were computed, and the relative quantification of mRNA expression was normalized against GAPDH and β-ACTIN employing the $2^{-\Delta\Delta Ct}$ quantification method. GVEA extract treatment resulted in a significant downregulation of EGFR and AKT1 mRNA expression (with fold changes

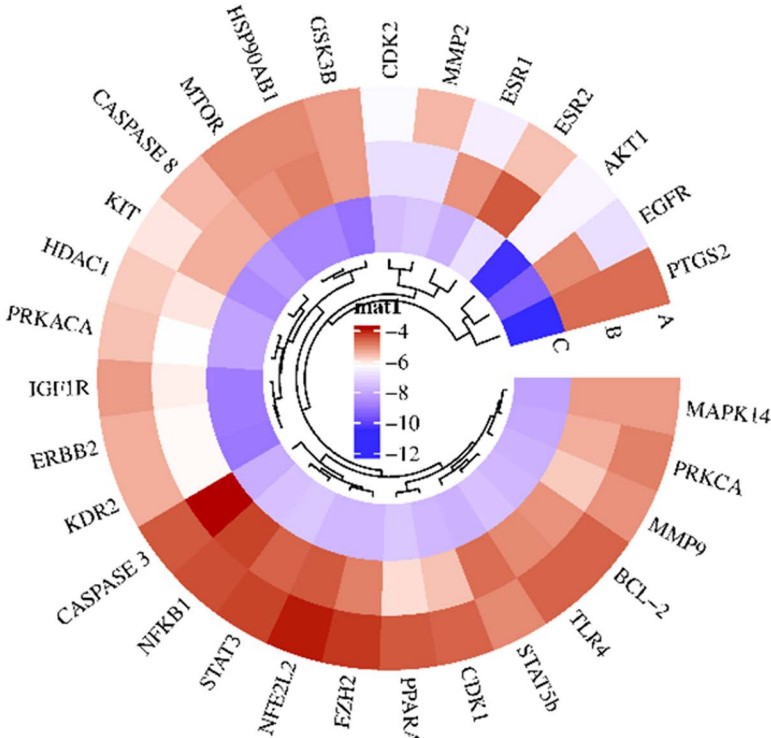

**Fig 6. Heat map showing molecular docking binding affinities.** Two GVEA extract compounds (**A**) 2,6,10-trimethyltetradecane, (**B**) Dodecan-2-ylbenzene, and (**C**) Doxorubicin hydrochloride (as our positive control) were evaluated for docking affinities against the top 30 hub genes.

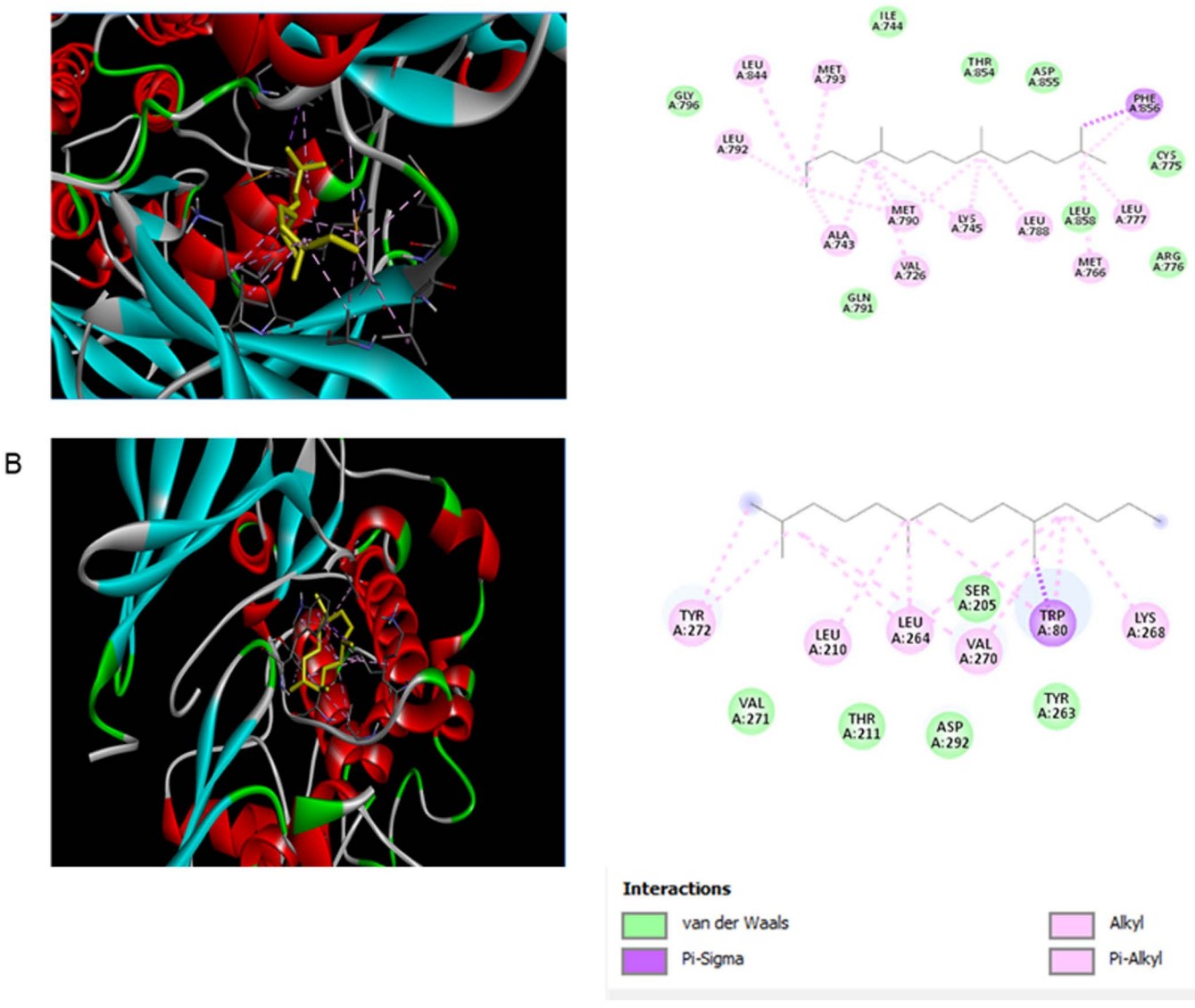

**Fig 7. Molecular Docking Illustrations.** Representative 3D and 2D diagrams showing the binding poses of 2,6,10-trimethyltetradecane, with (**A**) EGFR and (**B**) AKT1.

of 0.15±0.05 and 0.22±0.004, respectively) and a notable upregulation of TP53 and Caspase 3 expression (with fold changes of 2.57±0.38 and 2.78±0.29, respectively) (Fig 8).

## 4. Discussion

Cervical carcinoma represents the leading cause of mortality among females in Kenya, thereby constituting a major public health challenge. The diagnosis of cervical cancer in Kenya often occurs at advanced cancer stages, resulting in unfavorable prognostic outcomes. Additionally, the phenomenon of chemoresistance tends to happen more often when

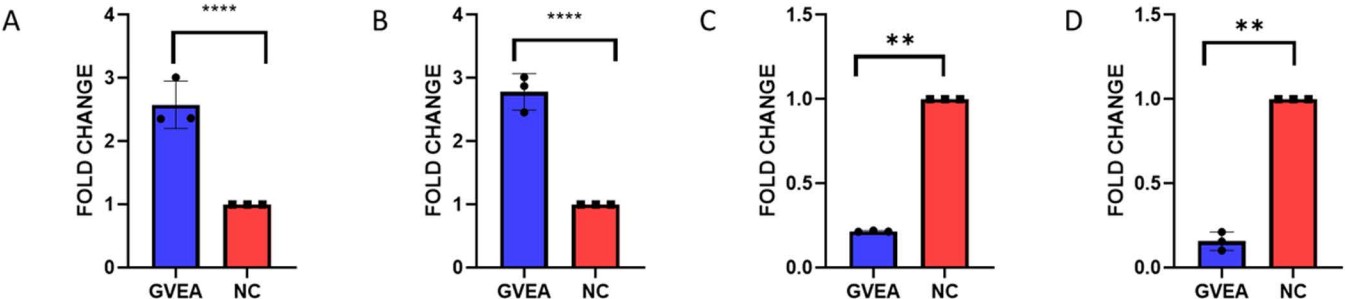

**Fig 8. RT-qPCR mRNA expression analysis.** HeLa cells treated with GVEA extract (blue colour) and medium with 0.4% DMSO as negative control (NC; red in colour) were evaluated for mRNA expression analysis of **(A)** TP53; **(B)** CASP3; **(C)** EGFR, and **(D)** AKT1 by RT-qPCR relative to internal housekeeping gene control (GAPDH and ß-ACTIN). Three technical replicates were used. GVEA: G. villosa ethyl acetate extract; NC: negative control; p value for Fig 8A & 8B, ****=p<0.0001; Fig 8C, **=0.0033; Fig 8D, **=0.0056.

chemotherapy is initiated at the late stages of cervical cancer. Further, chemotherapy is expensive for the majority in LMICs, besides being associated with various adverse effects [41]. Therefore, prospecting and discovery of novel therapeutic interventions that exhibit enhanced efficacy, are cost-effective, and have reduced side effects are a critical and urgent need. *G. villosa* is traditionally used as medicine against cancer, dysentery, cholera, wounds, and sores [18,19,42]. Previous studies have identified the presence of alkaloids, flavonoids, and phenolics in *G. villosa* extracts [42,43]. Furthermore, there is scientific evidence of antibacterial and antioxidant activity *of G. villosa* extracts [42,43]. However, there is a gap in scientific validation of its folkoloric use in the treatment and management of cancer, and this is the gap this study seeks to fill using the cervical cancer model cell line (HeLa cells). Furthermore, we also aimed to identify the phytochemicals and compounds that can be partly associated with the bioactivity of its extract. Importantly, we also sought to provide a mechanistic explanation of the demonstrated bioactivity through an integrated approach that employed cell culture, computational, and gene expression methods.

First, we used the MTT assay to evaluate the antiproliferative activity of GVEA extracts. MTT assay constitutes a colorimetric method involving the transformation of the MTT dye, characterized by its yellow hue, into purple formazan crystals. The degree of purple coloration is directly proportional to the percentage of cellular viability [39]. The extracts of *G. villosa* were initially screened at a fixed concentration of 200 µg/mL, and the GVEA extract was prioritized as exhibiting biological activity (defined arbitrarily from our previous work as the extract that will inhibit proliferation by >50%) [36]. According to studies by Kuete et al., [44], for edible plant materials, an $IC_{50}$ < 50 µg/mL shows strong antiproliferative activity and moderate antiproliferative activity is demonstrated by $IC_{50}$ values between 50 and 200 µg/mL, while low antiproliferative activity is shown in $IC_{50}$ values between 200 and 1000, and $IC_{50}$ values above 1000 µg/mL are considered to have no antiproliferative activity [44]. Therefore, GVEA extract demonstrates moderate antiproliferative activity (100.70 µg/mL; Fig 1 and Table 2); results that are consistent with findings from related species from the same genus, albeit from different solvent extract, which showed that the *Grewia asiatica* methanolic extract inhibited the proliferation of HeLa cells with an $IC_{50}$ value of 177.8 µg/mL [20,21]. Furthermore, GVEA extract activity was within the acceptable toxicity threshold, as it had an SI > 1.0 [45–47].

The late stages of cancer evolution are characterized by metastasis, which are changes that confer on cancer cells the capacity to disseminate, seed, and proliferate in distant organs and tissues. Most cancer deaths are associated with metastasis, a feature also prominent in the late stages of cervical cancer [48,49]. Metastasis is indeed recognized as one of the hallmarks of cancer [50]. Multiple approaches are used to test for metastatic properties, including scratch assay, cell exclusion assay, scatter test, transwell migration assay, and chamber-based assays, among others [51]. In this study, we employed the wound-healing/scratch assay as an *in vitro* model to demonstrate the potential of our test GVEA extract

in limiting the migration of HeLa cancer cells. We chose this assay because it is an established, commonly used, simple method for assessing the basic features of cancer cell migration [51]. It involves the creation of an artificial wound/scratch on a confluent monolayer of cancer cells, and the wound closure happens as cells migrate to the cell-free region over time [52]. GVEA extract demonstrated a significant antimigration activity in both time and concentration-dependent manner (Fig 3). The $IC_{50}$ concentration exhibited the most pronounced inhibitory effect, whereas the $IC_{12.5}$ concentration exhibited minimal inhibition of HeLa cell migration.

Phytochemicals are secondary metabolites that are present in plants. Qualitatively, GVEA extract contained alkaloids, saponins, phenols, and tannins (Table 3). These classes of phytochemicals have been reported to have antiproliferative activities [53–56]. Our findings agree with previous studies where the presence of alkaloids, flavonoids, and phenolics was detected in *G. villosa* [42,43]. Nevertheless, in our case, we did not detect flavonoids, and we speculate that this could be because we tested a specific fraction (ethyl acetate fraction) rather than the total crude extract, as tested in previous studies. Importantly, phytochemicals in a plant tend to vary qualitatively and even quantitively based on various factors including genetic characteristics, environmental factors including climate, altitude and soil type; the phase in plants life history when collection took place, the processing of the plant sample after collection and existence of a distinct pheno-type of a particular species (chemical races) [57,58]. Gas chromatography coupled with mass spectrometry constitutes a sophisticated analytical technique employed for the separation and identification of various compounds derived from botanical extracts. Identification of the compounds in plant extracts is vital in drug discovery, not only in identifying poten-tial therapeutic compounds but also in inspiring synthesis or semi-synthesis of complex compounds that can be used for the treatment of various diseases, including cervical cancer [59]. From GC-MS analysis, fourteen compound peaks were identified, with the majority of the compounds being hydrocarbons, phenols, and esters (Table 4), which is expected, con-sidering GC-MS identifies mostly volatile and small compounds [60,61].

Cancer is considered a complex systemic disease that involves many diverse pathways in the body and therefore necessitates a polypharmacological approach. Single-target therapies are hindered by shortcomings, including a higher likelihood of resistance, toxicity, and low efficacy. Therefore, ideal chemotherapeutic drugs should have diverse targets. Plant extracts have this multitargeting capacity since they contain multiple small molecules in low concentration, which often work synergistically and/or additively by "choking" many molecular targets in multiple pathways. In theory, plant extracts function in a 'polypharmacology' paradigm, entailing the use of a single product against multiple targets, and this paradigm is touted as the game-changer in the treatment/management of complex diseases such as cancer [62,63]. This approach is associated with low toxicities and minimal side effects compared to conventional single-target therapies [64–66]. Furthermore, the drug discovery process is a slow process (taking a median of 7.3 years to develop a new drug product), that is expensive (for example, developing an oncology product was estimated to cost between \$765 million and \$4.6 billion in 2020) and it has very low success rates 2020 [67,68]. Using computationally aided approaches such as network pharmacology and molecular docking promises to spur the drug discovery process, making it faster, cheaper, and successful.

Network pharmacology provides information about the pathways that compounds in a plant extract are more likely to interact with, offering an avenue to identify potential new targets for complex diseases such as cervical cancer. This study examined the interactome of GVEA extract compounds to their specific molecular targets, and how they overlapped with targets associated with cervical cancer. Interestingly, we identified targets and pathways that are associated with can-cer evolution, development, and progression; specifically, central carbon metabolism in malignancy, resistance to EGFR tyrosine kinase inhibitors, apoptosis, and the PI3K-AKT signaling pathways. We also used the molecular docking method, a structure-based screening method, whereby the 3D structures of two prioritized compounds identified by GC-MS in GVEA extract (Table 4) were used to validate potential interactions with the top 30 hub gene targets as prioritized from network analysis (Fig 5C). All the proteins exhibited favorable binding affinities (≤−4.0 kcal/mol) with the prioritized two compounds. 2,6,10-trimethyltetradecane showed good binding affinities with KIT (−6.1), CDK2 (−6.5), ESR1 (−6.1), AKT1

(−6.6) and EGFR (−7), while Dodecan-2-ylbenzene showed good binding affinities with CDK2 (−7), MMP2 (−7), PRKACA (−6.4), PPARA (−6.), AKT1(−6.6), KDR (−6.3), IGFR (−6.2), HDAC1(−6.1) and ERBB2 (−6.3). Notably, we got interested in AKT1 and EGFR due to their role in cancer development [69,70].

Epidermal growth factor receptor (EGFR) constitutes a receptor protein characterized by tyrosine kinase activity, which is localized on the cellular membrane and initiates numerous signaling pathways that facilitate cellular survival, proliferation, and angiogenesis. EGFR overexpression in cervical cancer ranges from 9–90% depending on the disease stage and the study methodology. Therapies for cancer treatment have been developed against EGFR through monoclonal antibodies and inhibitors that target the EGFR domain [71,72]. EGFR emerged as one of the top 30 hub genes through network interaction analysis, and we putatively validated it as a GVEA extract target by molecular docking, whereby it had a robust binding affinity (−7.0 kcal/mol) with the 2,6,10-trimethyltetradecane compound. To functionally validate docking findings, we used RT-qPCR to check if the mRNA levels of EGFR would be deregulated upon treatment with GVEA extracts. Interestingly, we observed significant downregulation of EGFR mRNA levels (Fig 8C) upon GVEA extract treatment compared to treatment with negative control (0.4% DMSO; $p = 0.0033$). AKT1 is a serine tyrosine kinase, a part of the PI$_3$K signaling cascade that favors cell survival, proliferation, metabolism, and angiogenesis [73,74]. Mutations in the PI$_3$K signaling cascade are common, including mutations in AKT1 [75]. A study by Rashmi et al., [76] showed that AKT1 inhibitors increased cell death of cervical cancer cells through disrupting the mTOR pathway [76]. AKT1 was also identified in this study, through network interaction analysis, as one of the top 30 hub genes. We validated it as a GVEA extract compound target through molecular docking, and it gave high docking scores of −6.6 kcal/mol with both compounds. Further functional validation by RT-qPCR demonstrated that GVEA extract treatment led to downregulation of AKT mRNA levels compared to treatment with the negative control (Fig 8D; $p = 0.0033$). Furthermore, we tested two additional targets (TP53 and CASP3), proteins that have canonical roles in apoptosis, a critical hallmark in cancer development and progression [77–80]. Caspase 3, is usually induced in early apoptosis phase, while tumor protein 53 has diverse signaling roles that ultimately license or favor apoptosis and thus inhibiting proteins that would have otherwise, favored cell survival [81–83]. Interestingly, treatment of HeLa cells with GVEA extract led to significant upregulations of Tumor protein 53 and Caspase 3 mRNA levels compared to cells treated with negative control (Fig 8A–8B; $p < 0.0001$). Taken together, we have provided a sound scientific and mechanistic explanation of how *G. villosa* ethyl acetate extract mediates its antiproliferative activity against the HeLa cell line, a model for cervical cancer. These findings provide a solid foundation for further drug discovery work on GVEA extract against cervical cancer.

## 5. Conclusions

This study provides solid evidence regarding the antiproliferative efficacy of *G. villosa* ethyl acetate extract against HeLa cell line, an *in vitro* model for cervical cancer. GVEA extract exhibited selective antiproliferative effects and suppressed cellular migration. We also identified phytochemicals and compounds that can be partly or wholly associated with the demonstrated activity. Further, we predicted putative molecular targets and pathways and validated these targets using molecular docking. Additionally, GVEA extract treatment modulated the expression of critical genes canonically associated with cancer development, progression, and apoptosis (including EGFR, AKT1, TP53, and Caspase 3). Importantly, 3 of these genes (EGFR, AKT1 and CASP3) were identified as part of top 30 hub genes and validated by docking studies. This demonstrates the usefulness of our integrated approach in validating the specificity of the reported activity. Therefore, our exploratory findings provide a knowledge-based basis for further drug discovery work on GVEA extract against cervical cancer.

## 6. Limitations and recommendations

This exploratory and hypothesis-generating study had a few limitations that create opportunities for future investigations. 1) We investigated total extracts and fractions, and characterized them by GC-MS. We recommend future characterization

of non-volatile and larger compounds in GVEA by LC-MS and subsequent isolation of pure compounds. 2) We investigated the antiproliferative activity of GVEA extract using the commonly used HeLa cell line. We recommend future incorporation of multiple cervical cancer cell lines, including SiHa, C33A, CaSki, and HaCaT cell lines. 3) We validated our docking results using gene expression analysis, a method that cannot account for post-translational modifications for some targets. We therefore recommend future validation of targets at the protein level using western blot methods. 4) We also used doxorubicin as our positive control, even though cisplatin is the widely used chemotherapeutic agent for the treatment of advanced cervical cancer. In the future, it is recommended to consider using cisplatin as an alternative to or in addition to doxorubicin as a comparator. Notably, both chemotherapeutic agents are used in the treatment of advanced cervical cancer, and a recent study has demonstrated that doxorubicin is more active on HeLa cells and more selective on Vero cells than cisplatin [84], suggesting its possible superiority as a comparator on these cell line models. We also recommend functional validation of apoptosis using Kits as well as *in vivo* validation of our findings using mouse model.

## Supporting information

**S1 File. S1 Fig. Cytotoxic effects of the experimental positive control.** We used doxorubicin hydrochloride as an experimental positive control. Doxorubicin hydrochloride was tested on Vero-ccl-81 cells at different concentrations to enable estimation of $CC_{50}$. NC: represent negative control (0.4% DMSO). **S2 Fig. Anti-proliferative effects of the positive control**. We used doxorubicin hydrochloride as an experimental positive control. Doxorubicin hydrochloride was tested on cancerous HeLa cells at different concentrations to enable estimation of $IC_{50}$. NC: represent negative control (0.4% DMSO).
(DOCX)

**S2 File. S1 Table. Grewia villosa compound targets.** S2 Table. Cervical cancer genes targets.
(XLSX)

## Acknowledgments

We express our gratitude to Kenyatta University, the Center for Traditional Medicine and Drug Research, and the Kenya Medical Research Institute (KEMRI) for their provision of essential resources and laboratory facilities. Furthermore, we extend our appreciation to Mr. Peter Maritim for his invaluable assistance and support.

## Author contributions

**Conceptualization:** Sally Wambui Kamau, Sospeter Ngoci Njeru.

**Data curation:** Sally Wambui Kamau, Sospeter Ngoci Njeru.

**Formal analysis:** Sally Wambui Kamau.

**Funding acquisition:** Sospeter Ngoci Njeru.

**Investigation:** Sally Wambui Kamau, Mercy Jepkorir, Gilbert Kipkoech, Inyani John Lino Lagu, Wesley Kanda, Susan Kibunja, Rakita Letoluo, Shadrack Barmasai, Alice Wanyoko, Vincent Ruttoh, James Kuria, Peter Githaiga Mwitari, Sospeter Ngoci Njeru.

**Methodology:** Sally Wambui Kamau, Mercy Jepkorir, Gilbert Kipkoech, Inyani John Lino Lagu, Wesley Kanda, Susan Kibunja, Rakita Letoluo, Shadrack Barmasai, Alice Wanyoko, Vincent Ruttoh, James Kuria, Peter Githaiga Mwitari, Sospeter Ngoci Njeru.

**Project administration:** Peter Githaiga Mwitari, Mathew Piero Ngugi, Sospeter Ngoci Njeru.

**Resources:** Peter Githaiga Mwitari, Sospeter Ngoci Njeru.

**Supervision:** James Kuria, Peter Githaiga Mwitari, Mathew Piero Ngugi, Sospeter Ngoci Njeru.

**Validation:** Sally Wambui Kamau, Peter Githaiga Mwitari, Mathew Piero Ngugi, Sospeter Ngoci Njeru.

**Visualization:** Sally Wambui Kamau, Peter Githaiga Mwitari, Mathew Piero Ngugi, Sospeter Ngoci Njeru.

**Writing – original draft:** Sally Wambui Kamau, Sospeter Ngoci Njeru.

**Writing – review & editing:** Sally Wambui Kamau, Mercy Jepkorir, Gilbert Kipkoech, Inyani John Lino Lagu, Wesley Kanda, Susan Kibunja, Rakita Letoluo, Shadrack Barmasai, Alice Wanyoko, Vincent Ruttoh, James Kuria, Peter Githaiga Mwitari, Mathew Piero Ngugi, Sospeter Ngoci Njeru.

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
