## [Decision Letter · Decision Letter 0]

5 Jun 2025

PONE-D-25-21556Antiproliferative Activity of Grewia villosa Ethyl Acetate Extract on Cervical Cancer HeLa Cell lines: Mechanistic Insights through Network Pharmacology and Functional Assays Approach

PLOS ONE

Comments PONE-D-25-21556

Dear Dr. Njeru,

Thank you for submitting your manuscript, Antiproliferative Activity of Grewia villosa Ethyl Acetate Extract on Cervical Cancer HeLa Cell lines: Mechanistic Insights through Network Pharmacology and Functional, to *PLOS ONE* . We appreciate the opportunity to consider your work and value the time and effort you have invested in your research.

After careful evaluation by our editorial team and peer reviewers, we have determined that your manuscript has the potential for publication in *PLOS ONE* after revisions. The reviewers have provided constructive feedback to help strengthen your paper, and their comments are included below for your reference.

While we recognize that revisions may require additional work, we hope you find the reviewers’ suggestions valuable in refining your manuscript. We invite you to submit a revised version addressing these concerns, along with a detailed point-by-point response to each comment.

We look forward to receiving your revised manuscript.

Kind regards,

Muhammad Zeeshan Bhatti, Ph.D

Academic Editor

PLOS ONE

 [This research partly received support from the KEMRI Internal Research Grant (IRG) funding (KEMRI/IRG/EC0017) to SNN.]. 

3. In the online submission form, you indicated that [All the data and information underlying results presented in this study are available and incorporated within the manuscript and the accompanying supplementary materials provided].

4. Please ensure that you refer to Figure 4 in your text as, if accepted, production will need this reference to link the reader to the figure.

5. Please include a caption for figure 5.

Reviewers' comments:

Reviewer's Responses to Questions

**Comments to the Author**

1. Is the manuscript technically sound, and do the data support the conclusions?

Reviewer #1: Yes

Reviewer #2: Yes

2. Has the statistical analysis been performed appropriately and rigorously? 

Reviewer #1: Yes

Reviewer #2: Yes

3. Have the authors made all data underlying the findings in their manuscript fully available?

Reviewer #1: Yes

Reviewer #2: Yes

4. Is the manuscript presented in an intelligible fashion and written in standard English?

Reviewer #1: Yes

Reviewer #2: Yes

5. Review Comments to the Author

Reviewer #1: Review

Title Antiproliferative Activity of Grewia villosa Ethyl Acetate Extract on Cervical Cancer HeLa Cell lines: Mechanistic Insights through Network Pharmacology and Functional Assays Approach the objectives and results reflect the title of the article

Introduction

1. It is necessary to rephrase the expression that chemotherapy is the standard of therapy for cervical cancer. It is necessary to clarify the standard for advanced stages and provide a link (line 62-63)

2. there is an indication of chemotherapy, it is necessary to indicate the importance of immunotherapy in the treatment of cervical cancer

3. The authors indicated the plant G. Villosa effectiveness in diseases unrelated to cancer (line 76-79)

Methods

1. Authors presented clearly and detailed information on methodology chapter

2. Safety and technique presented in the chapter

Results

1. Data performed chronologically

2. Data on chapter and in figures/tables equal

3. There is no interpretation of the results

Discussion

1. Chapter is not started from concise statement summarizing the main findings of the study : “Cervical carcinoma represents the predominant cause of mortality among females within Kenya, thereby constituting a substantial public health challenge for the nation.”

2. data from similar or previous studies provided.

3. the chapter contains repetitions of information from the results chapter

4. Authors presented limitations of the study

5. highlighted statement of the study for future direction

Decision

Minor revision

Reviewer #2: The study is original and explores a previously unstudied species (Grewia villosa) against cervical cancer using both in vitro and in silico methods. The manuscript is generally well-organized and methodologically sound.

There are few weaknesses being profound throughout the study as stated below: (please explain in your rebuttal)

1. Lack of multiple cancer cell lines: Only HeLa cells were used. Broader anticancer potential needs testing in other lines (e.g., SiHa, C33A).

2. Single-dose scratch assay is insufficient to claim anti-metastatic properties.

3. Docking scores are relatively weak (>-6 kcal/mol in many cases). Most binding energies were between -3 to -5 kcal/mol, indicating poor affinity.

4. RT-qPCR gene expression was only conducted at IC50 dose, 48h. No dose- or time-dependence explored.

5. No evidence of post-translational effects or functional protein modulation (e.g., western blotting for EGFR or AKT1).

6. Authors claim “novel anti-cancer therapeutic potential” and “effective therapy” — not justified by IC₅₀, lack of compound isolation, or in vivo data.

7. Conflation of moderate in vitro inhibition with drug-likeness is scientifically misleading.

8. Numerous grammatical and typographical issues. Please check throughout the manuscript

9. Repetition of phrases like “this is the first report” and “robust validation” are overstated.

10. Full chemical names in GC-MS table should be IUPAC-compliant.

11. Redundant citations (Kamau et al., 2024) appear excessively.

12. Need to define thresholds (e.g., what is considered "good" ADMET prediction).

13. Use professional English editing (e.g., Grammarly Premium or institutional writing support).

14. Ensure all figures are clearly labeled and referenced in the main text.

15. Add abbreviations list and ensure consistent formatting of units (µg/mL, °C, etc.).

16. Report exact p-values, effect sizes, and confidence intervals.

17. Clarify biological replicates (n ≥ 3) and technical replicates in each experiment.

18. Provide statistical comparison between groups (e.g., ANOVA with Tukey post hoc, not just Dunnett).

While the manuscript presents an interesting preliminary finding and uses an integrative approach, the conclusions are overstated and not sufficiently supported by strong bioactivity, compound specificity, or validation data. The study should be reframed as exploratory and hypothesis-generating. Once these revisions are addressed, the manuscript would be considerably strengthened and could be reconsidered for publication.

6. PLOS authors have the option to publish the peer review history of their article (what does this mean? ). If published, this will include your full peer review and any attached files.

**Do you want your identity to be public for this peer review?** For information about this choice, including consent withdrawal, please see our Privacy Policy .

Reviewer #1: **Yes: ** Raikhan Bolatbekova

Reviewer #2: No

---

## [Author Response · Author response to Decision Letter 1]

25 Jun 2025

Response to Editor and Reviewers:

Editors’ Comment:

On the funding of this work, we state the following:

This research partly received support from the KEMRI Internal Research Grant (IRG) funding (KEMRI/IRG/EC0017) to SNN. However, the funder had no role in study design, data collection and analysis, decision to publish, or preparation of the manuscript.

EDITORS COMMENTS

COMMENTS (C:) RESPONSES/REBUTTAL (R:)

C: Please ensure that your manuscript meets PLOS ONE's style requirements, including those for file naming

R: We have amended our naming style to suit the journal’s requirements. We have also adhered to the PLOS ONE formatting style throughout the manuscript.

C: Please include this amended Role of Funder statement in your cover letter; we will change the online submission form on your behalf.

R: We have included the statement in the cover letter, “This research partly received support from the KEMRI Internal Research Grant (IRG) funding (KEMRI/IRG/EC0017) to SNN. However, the funder had no role in study design, data collection and analysis, decision to publish, or preparation of the manuscript.”

C: All PLOS journals now require all data underlying the findings described in their manuscript to be freely available to other researchers, either 1. In a public repository, 2. Within the manuscript itself, or 3. Uploaded as supplementary information.

R: All the data generated in this study are included in the manuscript and the associated supplementary materials, and are all included in this submission.

C: Please ensure that you refer to Figure 4 in your text as, if accepted, production will need this reference to link the reader to the figure.

R: We have scanned through the entire document, and all figures are now referenced in the text, including Fig 3 (which was Figure 4).

C: Please include a caption for figure 5 (now 4).

R: A caption has been included in page 14, lines 368-369 “Fig 4. GC-MS Chromatogram of compounds contained in the G. villosa ethyl acetate extract. Fourteen compound peaks were identified from G. villosa ethyl acetate extract fraction.

COMMENTS FROM REVEIWER #1

COMMENTS RESPONSES

C: It is necessary to rephrase the expression that chemotherapy is the standard of therapy for cervical cancer. It is necessary to clarify the standard for advanced stages and provide a link (line 62-63)

R: We acknowledge that there are many treatment options for cervical cancer and have introduced these options as well as clarified that chemotherapy is widely used in the late stages, which is the stage at which it is typically diagnosed in Kenya (Line 66-72), “The treatment of cervical cancer depends on the cancer stage and the extent of disease progression. The available treatment strategies may include one or a combination of surgery, radiation, immunotherapy, and chemotherapy [8, 9]. The use of chemotherapy is still the primary standard method for treating cervical cancer, especially in its late stages [10]. Nevertheless, chemotherapy poses significant challenges, including the evolution of chemotherapy resistance, severe adverse reactions, and high treatment costs that negatively impact patient quality of life and overall financial stability.”

C: There is an indication of chemotherapy, it is necessary to indicate the importance of immunotherapy in the treatment of cervical cancer

R: We acknowledge the importance of the various treatment options available and efficacious for cervical cancer. We have introduced these options including immunotherapy (line 66-72). However, since this study aimed at assessing the anti-proliferative activity of plant extracts, which contain potential bioactive compounds, we chose to amplify the chemotherapy treatment option, as opposed to immunotherapy and radiotherapy that as these two aspects cannot be compared at the same level with phytocompounds, but this comparison is possible with chemotherapy.

C: The authors indicated the plant G. Villosa effectiveness in diseases unrelated to cancer (line 76-79)

R: We added that section to highlight some of the traditional uses of the selected plant in different regions of the world; however, we agree that it does not add relevance to the disease being studied currently, and therefore, this part has been deleted.

C: Chapter is not started from concise statement summarizing the main findings of the study : “Cervical carcinoma represents the predominant cause of mortality among females within Kenya, thereby constituting a substantial public health challenge for the nation.”

R: The essence of this statement is to lay a foundation upon which our findings will be based, especially in light of the local context. We have rephrased this statement to bring out the message in unambiguous way, lines 496-500, “Cervical carcinoma represents the leading cause of mortality among females in Kenya, thereby constituting a major public health challenge. The diagnosis of cervical cancer in Kenya often occurs at advanced cancer stages, resulting in unfavorable prognostic outcomes. Additionally, the phenomenon of chemoresistance tends to happen more often when chemotherapy is initiated at the late stages of cervical cancer.”

C: the chapter contains repetitions of information from the results chapter

R: The repetitions have been removed

COMMENTS FROM REVIEWER #2

COMMENTS RESPONSES

C: Lack of multiple cancer cell lines: Only HeLa cells were used. Broader anticancer potential needs testing in other lines (e.g., SiHa, C33A).

R: We acknowledge that multiple cell lines would have further strengthened our data on cervical cancer. However, we used the only available cell line for cervical cancer (HeLa cell line). Notably, HeLa cell lines is the most commonly, and widely available model for cervical cancer research. We do not have these additional cell lines stocked in the country, coupled with limited funding for this student project, and therefore we could not we could not afford additional cell lines due to cost limitation. However, our data robustness is reinforced in that there were at least three biological and three technical replicates for every experiment. Nevertheless, we have captured this a project limitation, and recommended additional cell lines for fure work, see lines 642-645 “We investigated the antiproliferative activity of GVEA extract using the commonly used HeLa cell line. We recommend future incorporation of multiple cervical cancer cell lines, including SiHa, C33A, CaSki, and HaCaT cell lines.”..

C: Single-dose scratch assay is insufficient to claim anti-metastatic properties.

R: We acknowledge that metastasis is a complex cellular phenomenon that cannot be explained with one test. We wanted to demonstrate the capacity of our extracts to limit cellular migration, and we have clarified this in the text, by modifying texts that suggests as if we wanted to test metastasis. We have clarified that multiple tests are required to answer the metastasis question, and that in this study we limited ourselves to testing cell migration using the commonly used in vitro model; see lines 529-539

“The late stages of cancer evolution are characterized by metastasis, which are changes that confer on cancer cells the capacity to disseminate, seed, and proliferate in distant organs and tissues. Most cancer deaths are associated with metastasis, a feature also prominent in the late stages of cervical cancer [47, 48]. Metastasis is indeed recognized as one of the hallmarks of cancer [49]. Multiple approaches are used to test for metastatic properties, including scratch assay, cell exclusion assay, scatter test, transwell migration assay, and chamber-based assays, among others [50]. In this study, we employed the wound-healing/scratch assay as an in vitro model to demonstrate the potential of our test GVEA extract in limiting the migration of HeLa cancer cells. We chose this assay because it is an established, commonly used, simple method for assessing the basic features of cancer cell migration [50]. It involves the creation of an artificial wound/scratch on a confluent monolayer of cancer cells, and the wound closure happens as cells migrate to the cell-free region over time [51].”

C: Docking scores are relatively weak (>-6 kcal/mol in many cases). Most binding energies were between -3 to -5 kcal/mol, indicating poor affinity.

R: We acknowledge the weak binding affinities of most of our compounds with the protein targets.

The thresholds used to explain our findings are explained in lines 449-453 “A binding energy value of less than 0 kcal/mol signifies that a ligand and a receptor exhibit spontaneous binding. However, a binding affinity of less than -5.0 kcal/mol denotes a strong binding affinity between the ligand and the receptor, while a binding affinity of less than -7.0 kcal/mol indicates an exceptionally strong binding affinity between the ligand and the receptor [39]. Our 3 tested compounds exhibited a binding affinity of ≤ -4 kcal/mol (Fig 6),” and lines 588-593, “All the proteins exhibited favorable binding affinities (≤-4.0 kcal/mol) with the prioritized two compounds. 2,6,10-trimethyltetradecane showed good binding affinities with KIT (-6.1), CDK2 (-6.5), ESR1 (-6.1), AKT1 (-6.6) and EGFR (-7), while Dodecan-2-ylbenzene showed good binding affinities with CDK2 (-7), MMP2 (-7), PRKACA (-6.4), PPARA (-6.), AKT1(-6.6), KDR (-6.3), IGFR (-6.2), HDAC1(-6.1) and ERBB2 (-6.3). Notably, we got interested in AKT1 and EGFR due to their role in cancer development [68, 69]”. Importantly, we used threshold of <-5kcal/mol to prioritize the targets selected for validation by RT-qPCR, including AKT1 and EGFR.

C: RT-qPCR gene expression was only conducted at IC50 dose, 48h. No dose- or time-dependence explored.

R: We acknowledge that conducting the RT-qPCR analysis at multiple doses would have possibly provided more results, though explaining the same aspect. Based on our functional assays, namely the morphological analysis, antimigration assay, treatment with IC50 concentrations yielded the highest results, and results obeyed concentration kinetic dynamics. It was therefore reasonable to pick the same value and expect optimal deregulation of target genes. Also, the cost of Kits involved for this experiment are pricy and we wanted to test a couple of targets, and therefore we went for the optimal concentration and optimal time.

C: No evidence of post-translational effects or functional protein modulation (e.g., western blotting for EGFR or AKT1)

R: We agree that testing for post-translation modification by western blot approach would have added another layer of information to our findings. Unfortunately, the cost implication for the reagents and infrastructure to do western blot are behold the minimal funding for this study. Nevertheless, we pride ourselves to have used integrated approach to identify these targets, validate them through docking and functionally validate them through gene expression analysis. We have captured the inability to test post-translational modification as a limitation in this study, see lines 643-647, “We validated our docking results using gene expression analysis, a method that cannot account for post-translational modifications for some targets. We therefore recommend future validation of targets at the protein level using western blot methods.”.

C: Authors claim “novel anti-cancer therapeutic potential” and “effective therapy” — not justified by IC₅₀, lack of compound isolation, or in vivo data.

R: We acknowledge the possibilities of these statements sounding to over claim our finding. We have modified by replacing the statement with effective therapy, to read in lines 49-50 “This highlights the plant's potential in discovering products and compounds for further investigation on possible application in cervical cancer management and/or treatment.”

Lines 501-503 “Therefore, prospecting and discovery of novel therapeutic interventions that exhibit enhanced efficacy, are cost-effective, and have reduced side effects are a critical and urgent need.” is a general statement that lays the foundation to our study contribution to global need, and its not being given to suggest our findings. Therefore I take this as a misunderstanding, but we have rephrased it to bring the intended background easily.

C: Conflation of moderate in vitro inhibition with drug-likeness is scientifically misleading.

R: We acknowledge that this statement in its form risk misleading and we have restructured it to now read in lines 584-588, “We also used the molecular docking method, a structure-based screening method, whereby the 3D structures of two prioritized compounds identified by GC-MS in GVEA extract (Table 4) were used to validate potential interactions with the top 30 hub gene targets as prioritized from network analysis (Fig 5C).”

C: Numerous grammatical and typographical issues. Please check throughout the manuscript

R: We have gone through the entire manuscript and made the relevant changes to fix all typos.

C: Repetition of phrases like “this is the first report” and “robust validation” are overstated.

R: We have removed repetitive words and phrases.

C: Full chemical names in GC-MS table should be IUPAC-compliant.

R: We have changed all the compound names to be compliant with the IUPAC naming system

C: Redundant citations (Kamau et al., 2024) appear excessively.

R: We have solved this by removing redundancy in citations.

C: Need to define thresholds (e.g., what is considered "good" ADMET prediction).

R: We have included the threshold values to help in interpreting ADMET predictions as shown in lines 202-215 “The parameters obtained from these analytical platforms were utilized to predict the drug-likeness of the compounds in accordance with Lipinski's rules, while applying the following threshold; there should be no more than 5 H-bond donors, there should be less than 10 H-bond acceptors, the molecular weight should not be greater than 500, the calculated LogP should not be greater than 5, and the number of rotatable bonds in a compound should be less than 10. Additionally, compounds’ good absorption properties were predicted through indices, including the topological polar surface area (TPSA) threshold of <140 Å2), water solubility threshold in log mol/L of >-6), human intestinal absorption threshold of >30%, and Caco-2 permeability threshold in log Papp (10-6 cm/s) of >0.90). Putative good distribution properties were predicted through features including blood‒brain barrier (BBB) penetration threshold of logBB >0.3, and central nervous system (CNS) penetration threshold of logPS >-2. Total clearance was predicted in log mL/min/kg, and the higher the value, the better, while the acceptable predicted threshold for the maximum recommended tolerated dose in humans (MRTD) is ≤ 0.477 log(mg/kg/day) [31-34].”

C: Use professional English editing (e.g., Grammarly Premium or institutional writing support).

R: We have used Grammarly premium to run through the document.

C: Ensure all figures are clearly labeled and referenced in the main text.

R: We have ensured that all figures are well labelled and referenced in the main text as advised.

We have also deleted one figure that we considered redundant as its information is already showed in table 2

C: Add abbreviations list and ensure consistent formatting of units (µg/mL, °C, etc.).

R: We have added an abbreviation list and made the appropriate corrections for the consistency needed

C: Report exact p-values, effect sizes, and confidence intervals.

R: We have given the exact p-values in the legend by explaining the meaning of the stars affixed in the images. Putting the exact p-values in the image may unnecessarily cloud the figures and the reason for putting them in the legend.

C: Clarify biological replicates (n ≥ 3) and technical replicates in each experiment.

R: We have clarified technical and experimental replicates for each experiment.

C: Provide statistical comparison between groups (e.g., ANOVA with Tukey post hoc, not just Dunnett). ANOVA with Tukey post hoc test as well as Dunnetts test was used for all analysis using Graphpad 8.0 version.

R: This

---

## [Decision Letter · Decision Letter 1]

7 Aug 2025

PONE-D-25-21556R1Antiproliferative Activity of Grewia villosa Ethyl Acetate Extract on Cervical Cancer HeLa Cell lines: Mechanistic Insights through Network Pharmacology and Functional Assays ApproachPLOS ONE

Dear Dr. Njeru,

Thank you for submitting your manuscript to PLOS ONE. After careful consideration, we feel that it has merit but does not fully meet PLOS ONE’s publication criteria as it currently stands. Therefore, we invite you to submit a revised version of the manuscript that addresses the points raised during the review process.

We look forward to receiving your revised manuscript.

Kind regards,

Muhammad Zeeshan Bhatti, Ph.D

Academic Editor

PLOS ONE

Journal Requirements:

Additional Editor Comments:

The reviewer raised some minor questions that need to be addressed.

Reviewers' comments:

Reviewer's Responses to Questions

**Comments to the Author**

1. If the authors have adequately addressed your comments raised in a previous round of review and you feel that this manuscript is now acceptable for publication, you may indicate that here to bypass the “Comments to the Author” section, enter your conflict of interest statement in the “Confidential to Editor” section, and submit your "Accept" recommendation.

Reviewer #1: All comments have been addressed

Reviewer #2: All comments have been addressed

2. Is the manuscript technically sound, and do the data support the conclusions?

Reviewer #1: Yes

Reviewer #2: Yes

3. Has the statistical analysis been performed appropriately and rigorously? 

Reviewer #1: Yes

Reviewer #2: Yes

4. Have the authors made all data underlying the findings in their manuscript fully available?

Reviewer #1: Yes

Reviewer #2: Yes

5. Is the manuscript presented in an intelligible fashion and written in standard English?

Reviewer #1: Yes

Reviewer #2: Yes

6. Review Comments to the Author

Reviewer #1: Review

Title

Antiproliferative Activity of Grewia villosa Ethyl Acetate Extract on Cervical Cancer HeLa Cell lines: Mechanistic Insights through Network Pharmacology and Functional Assays Approach

Abstract

1. Authors were indicated that this study is a first study which reported antiprolifirated activity of Grewia villosa in cervical cancer. It is recommended to write that knowledge is limited or absent. It is impossible to be sure that the research is unique.

Introduction

Chemotherapy is not primary treatment of cervical cancer. Chemotherapy alone recommended as a primary treatment in advanced stage of cervical cancer without PDl mutation

Methods

Chapter methods clearly written

Line 190 Recommended to clarify of photochemical screening test of GVEA

There is no presented information on Ethics

Results

Data performed chronologically

Data on chapter and in figures/tables equal

There is an interpretation of the results in all sub chapters

Line 284 Why authors decided to examine doxorubicine in control group. Why cisplatin is not used for control group as a chemotherapy drug choice according to guidelines

Discussion

1. Chapter is not started from concise statement summarizing the main findings of the study

2. Strengths and weaknesses of study are presented clearly in chapter

3. Highlighted statement what is offered by the study for future direction

Reviewer #2: (No Response)

7. PLOS authors have the option to publish the peer review history of their article (what does this mean? ). If published, this will include your full peer review and any attached files.

**Do you want your identity to be public for this peer review?** For information about this choice, including consent withdrawal, please see our Privacy Policy .

Reviewer #1: **Yes: ** Raikhan Bolatbekova

Reviewer #2: No

---

## [Author Response · Author response to Decision Letter 2]

11 Aug 2025

Editor: Thank you for submitting your manuscript to PLOS ONE. After careful consideration, we feel that it has merit but does not fully meet PLOS ONE’s publication criteria as it currently stands. Therefore, we invite you to submit a revised version of the manuscript that addresses the points raised during the review process

Response: Thank you, we have addressed the concerns raised by the reviewer.

Editor: Please include the following items when submitting your revised manuscript:

Response: All these have been done

Editor: If the reviewer comments include a recommendation to cite specific previously published works, please review and evaluate these publications to determine whether they are relevant and should be cited. There is no requirement to cite these works unless the editor has indicated otherwise.

Response: NA

Editor: Please review your reference list to ensure that it is complete and correct. If you have cited papers that have been retracted, please include the rationale for doing so in the manuscript text, or remove these references and replace them with relevant current references. Any changes to the reference list should be mentioned in the rebuttal letter that accompanies your revised manuscript. If you need to cite a retracted article, indicate the article’s retracted status in the References list and also include a citation and full reference for the retraction notice.

Response: References checked and are all fine

Editor: The reviewer raised some minor questions that need to be addressed.

Response: I have addressed them all as shown below

COMMENTS FROM REVIEWER

1. Abstract

Authors were indicated that this study is a first study which reported antiprolifirated activity of Grewia villosa in cervical cancer. It is recommended to write that knowledge is limited or absent. It is impossible to be sure that the research is unique.

Response: This was captured in the tracked copy before corrections were made, but was already rectified in the clean copy. I have highlighted (from line 45 and 87) how it reads, and we have avoided the mention of being first study throughout the document. On line 87, I have modified the sentence to read “Given the lack of scientific data… ”

2. Introduction

Chemotherapy is not primary treatment of cervical cancer. Chemotherapy alone recommended as a primary treatment in advanced stage of cervical cancer without PDl mutation

Response: We have corrected this and appended a new different reference and now reads “Chemotherapy is usually used as an integral part of the standard cervical cancer treatment regimen, either as an adjuvant therapy following surgery to forestall recurrence, in combination with radiotherapy, or as a standalone therapy for locally advanced cervical cancer without PDl mutation [10]”…line 68

3. Methods

Chapter methods clearly written

3a) Line 190 Recommended to clarify of photochemical screening test of GVEA

3b) There is no presented information on Ethics

Response 3a): We have added more information to make this section more clear, and now reads “The qualitative phytochemical screening tests of GVEA extract were conducted, and classes of phytochemicals were identified through characteristic color changes according to standard methods. Particularly, sulfuric acid solution and Mayer’s reagent test were used to identify alkaloids, water frothing and olive oil emulsion test were used to identify saponins, alkaline reagent (aqueous ammonia solution) test was used to identify flavonoids, Salkowski test was used to identify terpenoids, Keller-Killiani test was used to identify glycosides, ferric chloride and lead acetate test were used to identify tannins, and sodium hydroxide and ferric chloride tests were used to identify phenolics [28-30].”

Response 3b): According to the Plos One Journal instructions, the information on ethics is usually captured in the online submission system and not on the attached document. We have already captured the ethics statement in the online submission system, and it reads as “Approval for this study was granted by the Scientific Ethical Review Unit of the Kenya Medical Research Institute (KEMRI/SERU/CTMDR/104/4466)”

4 Results

Data performed chronologically

Data on chapter and in figures/tables equal

There is an interpretation of the results in all sub chapters

Line 284 Why authors decided to examine doxorubicine in control group. Why cisplatin is not used for control group as a chemotherapy drug choice according to guidelines

Response: Thank you for compliments.

The reviewer is correct that Cisplatin is a widely used chemotherapeutic drug for cervical cancer; however, the literature also indicates that cisplatin and doxorubicin are the two most commonly used drugs (https://dx.doi.org/10.13005/bpj/3133). This publication has recently demonstrated that doxorubicin exhibits higher activity and selectivity than cisplatin, indicating that it provides a more suitable comparator. Secondly, doxorubicin was the control available in our labs. We will capture this aspect as a limitation of the study, see line 656 “4) We also used doxorubicin as our positive control, even though cisplatin is the widely used chemotherapeutic agent for the treatment of advanced cervical cancer. In the future, it is recommended to consider using cisplatin as an alternative to or in addition to doxorubicin as a comparator. Notably, both chemotherapeutic agents are used in the treatment of advanced cervical cancer, and a recent study has demonstrated that doxorubicin is more active on HeLa cells and more selective on Vero cells than cisplatin [84], suggesting its possible superiority as a comparator on these cell line models.”.

5 Discussion

1. Chapter is not started from concise statement summarizing the main findings of the study

2. Strengths and weaknesses of study are presented clearly in chapter

3. Highlighted statement what is offered by the study for future direction

Response 5.1: We chose a different approach of starting the section by laying the foundation, giving the finding and ending the section with the concise summary of the main finding. This is considering we have offered the same at the end of abstract, introduction and as part of our conclusing.

Response 5.2: Thank you for positive affirmation

Response5.3 : Thank you for positive affirmation

---

## [Editor Report · Decision Letter 2]

20 Aug 2025

Antiproliferative Activity of Grewia villosa Ethyl Acetate Extract on Cervical Cancer HeLa Cell lines: Mechanistic Insights through Network Pharmacology and Functional Assays Approach

PONE-D-25-21556R2

Dear Dr. Njeru,

We’re pleased to inform you that your manuscript has been judged scientifically suitable for publication and will be formally accepted for publication once it meets all outstanding technical requirements.

Kind regards,

Muhammad Zeeshan Bhatti, Ph.D

Academic Editor

PLOS ONE
---

## [Editor Report · Acceptance letter]

PONE-D-25-21556R2

PLOS ONE

Dear Dr. Njeru,

I'm pleased to inform you that your manuscript has been deemed suitable for publication in PLOS ONE. Congratulations! Your manuscript is now being handed over to our production team.

Kind regards,

on behalf of

Dr. Muhammad Zeeshan Bhatti

Academic Editor

PLOS ONE